# Filovirus Neutralising Antibodies: Mechanisms of Action and Therapeutic Application

**DOI:** 10.3390/pathogens10091201

**Published:** 2021-09-16

**Authors:** Alexander Hargreaves, Caolann Brady, Jack Mellors, Tom Tipton, Miles W. Carroll, Stephanie Longet

**Affiliations:** 1Nuffield Department of Medicine, Wellcome Centre for Human Genetics, University of Oxford, Oxford OX3 7BN, UK; Alexander.Hargreaves@well.ox.ac.uk (A.H.); Caolann.Brady@well.ox.ac.uk (C.B.); Jack.Mellors@phe.gov.uk (J.M.); Tom.Tipton@well.ox.ac.uk (T.T.); miles.carroll@ndm.ox.ac.uk (M.W.C.); 2Faculty of Health and Medical Sciences, University of Surrey, Guildford GU2 7XH, UK; 3National Infection Service, Public Health England, Porton Down, Salisbury SP4 0JG, UK; 4Department of Infection Biology, Institute of Infection and Global Health, University of Liverpool, Liverpool L69 7ZX, UK

**Keywords:** neutralising antibodies, filoviruses, ebolavirus, post-vaccination, post-infection, monoclonal antibodies, longitudinal antibody response

## Abstract

Filoviruses, especially Ebola virus, cause sporadic outbreaks of viral haemorrhagic fever with very high case fatality rates in Africa. The 2013–2016 Ebola epidemic in West Africa provided large survivor cohorts spurring a large number of human studies which showed that specific neutralising antibodies played a key role in protection following a natural Ebola virus infection, as part of the overall humoral response and in conjunction with the cellular adaptive response. This review will discuss the studies in survivors and animal models which described protective neutralising antibody response. Their mechanisms of action will be detailed. Furthermore, the importance of neutralising antibodies in antibody-based therapeutics and in vaccine-induced responses will be explained, as well as the strategies to avoid immune escape from neutralising antibodies. Understanding the neutralising antibody response in the context of filoviruses is crucial to furthering our understanding of virus structure and function, in addition to improving current vaccines & antibody-based therapeutics.

## 1. Filoviridae Background

### 1.1. Filoviridae Phylogeny

The first filovirus genus to be identified was *Marburgvirus* in 1967 composed of one species *Marburg marburgvirus* with two viruses: Marburg virus (MARV) and the very closely related Ravn virus (RAVV). Both viruses cause Marburg virus disease (MVD), a highly lethal form of viral haemorrhagic fever, with the largest outbreak occurring between 2004 and 2005 in Angola, with 252 infected individuals and a case fatality rate (CFR) of 90% [1].

The second and most notorious genus of filovirus, *Ebolavirus*, contains six species each with one virus; *Zaire ebolavirus*, Ebola virus (EBOV); *Sudan ebolavirus*, Sudan virus (SUDV); *Bundibugyo ebolavirus*, Bundibugyo virus (BDBV); *Tai Forest ebolavirus*, Tai Forest virus (TAFV); *Reston ebolavirus*, Reston virus (RESTV) and *Bombali ebolavirus*, Bombali virus (BOMV). The first four of these six viruses are known to cause Ebola virus disease (EVD) in humans, with a CFR frequently reported between 40% and 90%. However, this is likely an overestimate as many EBOV infections may go unreported [2]. EBOV is predominantly responsible for the EVD outbreaks of the greatest magnitude, with the largest being the 2013–2016 West African outbreak [3,4].

Most recently, new filoviruses have been discovered, which have not yet been associated with outbreaks in humans. In 2011, a third genus *Cuevavirus* was discovered with a sole species *Lloviu cuevavirus* including one virus, Lloviu virus (LLOV) [5]. Whilst infectious LLOV still remains to be isolated, anti-LLOV antibodies have been detected in bats [6,7]. In 2019, a new filovirus species *Měnglà dianlovirus*, including Mengla virus (MLAV) was discovered in China as the sole species of a new genus *Dianlovirus* [8]. Still to date, neither BOMV, LLOV nor MLAV are known to cause viral haemorrhagic fever with its pathogenicity in humans still to be determined [8,9]. However, both BOMV and MLAV Glycoprotein (GP) pseudoviruses as well as LLOV virus-like particles (VLPs) demonstrate a broad tissue tropism in cell lines from different animals, replicate similarly to ebolaviruses and use the Niemann pick type C1 (NPC1) as an entry receptor indicating a potential for spillover events [3,8,10]. Two much more divergent species of filovirus *Huángjiāo thamnovirus* including Huángjiāo virus (HUJV) and *Xīl**ǎng striavirus* including Xīlǎng virus (XILV), which belong to the genera *Thamnovirus* and *Striavirus* respectively, have been also described. These filoviruses infect fish [11].

While non-EBOV filoviruses will be mentioned, this review will be more focused on EBOV due to the ongoing impact of this pathogen on world health and the recent developments in antibody-based therapeutics and EBOV vaccines.

### 1.2. Genome Organisation of Ebolavirus

With the exception of the more divergent *Thamnovirus* and *Striavirus* genera, all filoviruses encode seven structural proteins: Nucleoprotein (NP), Viral Protein (VP) VP35, VP40, GP, VP30, VP24 and L polymerase (L) as shown in Figure 1A.

The NP is the main component of the ribonucleoprotein or nucleocapsid, though VP30 and VP24 are also required for the stability of the nucleocapsid and together make the changes in NP required to incorporate the viral RNA. L polymerase is an RNA-dependent RNA polymerase which complexes with the polymerase co-factor VP35 and is responsible for transcribing the viral RNA, while the initiation of transcription is activated by VP30. L polymerase also functions in regulating and editing the viral RNA e.g., in the case of GP where three different gene products are produced [14]. VP40 is considered the matrix protein and is crucial for viral assembly and budding. It is also worth noting that many of the viral proteins, particularly VP24, VP30, VP35 and VP40 have functions linked to host pathology or immune evasion. For instance, VP24 and VP35 inhibit interferon (IFN) pathways (reviewed in Cantoni and Rossman, 2018) [15].

### 1.3. Cellular Entry of Ebola Virus

Cell entry is a critical stage in the lifecycle of any virus. GP exposed on the surface of the virion has been demonstrated to be essential for cell entry. In which case GP is the most important target for neutralising antibodies.

In EBOV, co- or post- translational editing (e.g., transcriptional slippage) of the GP transcript produce various GP products (reviewed in detail by Lee et al.) [12]. Similar post-translational modifications are not observed in MARV (Figure 1A). In EBOV, the pre-GP gene is transcribed as one protein but is cleaved by the furin protease into two subunits, GP_1_ and GP_2_, joined by a disulphide bridge to form a heterodimer [16]. Surface GP exists as a trimer of the GP heterodimers expressed on the surface of the virion and interacts with NPC1 for cell entry [17]. However, surface GP is not the main gene product. The primary product, which is expressed from unedited RNA transcripts, is a non-structural protein called secreted GP (sGP). Secreted GP is a dimer of GP_1_ and a truncated GP_2_ bound by disulphide bridges [18] which is hypothesised to act as a decoy antigen to sequester antibodies or cause antigenic subversion. Surface GP is only produced upon the addition of an adenosine residue to the GP transcript in 20% of cases. A deletion of one adenosine or addition of two adenosines in the transcript leads to the production of another non-structural protein, small secreted GP (ssGP) [12,19]. The role of ssGP is unclear in EBOV pathogenesis. Lastly, during EBOV infection, GP can be shed from infected cells in a soluble form due to cleavage by TACE metalloprotease. This soluble GP is named shed GP and was shown to sequester EBOV-specific neutralising antibodies directed towards GP [20].

The entire GP is required for cell entry. GP_1_ is associated with cell attachment and receptor binding, whilst GP_2_ has functions regarding fusion with the host cell membrane. Both subunits are therefore targets of neutralising antibodies. GP_1_ contains four subdomains: the head, base, mucin-like domain (MLD) and glycan cap (GC). The GP_1_ head, when arranged in its trimeric conformation forms a three lobed chalice containing the receptor binding domain (RBD). The GP_1_ base clamps the internal fusion loop (IFL) and heptad repeat 1 (HR1) and heptad repeat 2 (HR2) from GP_2_ for arrangement into its pre-fusion conformation (Figure 1B) [12]. There are also two heavily glycosylated regions on the GP_1_, GC and the MLD [12]. The GC is a large chain of sugars, some of which are thought to act as attachment factors to host cells e.g., binding to C-type lectins. Both the MLD and GC are highly variable regions and can block neutralising antibodies binding to the GP, e.g., the MLD overhangs the GP to protect critical epitopes [21].

GP_2_ is docked into the viral membrane by its transmembrane domain. GP_2_ is also responsible for fusing the virus membrane and the host endosome membrane. It contains several domains that are key to this process such as HR1 and HR2 that change conformation to aid the IFL insertion into the host endosome membrane ultimately leading to fusion (Figure 1B) [12].

EBOV uses a broad range of binding factors to attach to a variety of host cell types before being internalised by the host cell into an endocytic compartment by receptor mediated endocytosis or macropinocytosis [12,22]. In the late endosome, cathepsins cleave the GP removing the MLD and GC and exposing the RBD in a stage called priming, allowing better access to the RBD for receptor binding [17,23]. The RBD then binds to the NPC1 cholesterol transporter [24]. The base of GP_1_ is a clamp, that when released triggers the conformational changes required to expose the hydrophobic IFL which inserts into the endosomal membrane [12,25]. HR1 and HR2 pull on the IFL causing it to fuse with endosomal membrane, creating a pore for the release of the ribonucleoprotein into the cell cytoplasm [12,26].

### 1.4. EVD and Immunity

EVD is characterised as a systemic inflammatory response (“septic-shock-like-syndrome”) with coagulation abnormalities and multi-organ failure, immunosuppression and lymphopenia [4]. This pathology is a result of GP-mediated activation of innate immune cells.

EBOV preferentially targets dendritic cells (DC), monocytes and macrophages but can also productively infect epithelial and endothelial cells, adrenal fibroblasts and hepatocytes [27]. GP-mediated activation of the TLR4 pathway in DCs and macrophages [28,29] results in the expression and secretion of inflammatory cytokines (e.g., IL-6, IL-1β, TNF) [30,31,32,33]. In macrophages, EBOV elicits a strong upregulation of IFN-I signalling [34], a crucial feature of the immune response to viral infection that provides the link between innate and adaptive immunity [35]. However, the role of Type I IFN in EBOV clearance and protection is not fully understood. Several studies demonstrated that IFN-I impacted EBOV replication in vitro and survival in animal models [36,37,38,39,40], while higher levels of IFN-α were associated with EVD fatal cases [41]. With that said, the picture is likely more complicated as EBOV has also developed some strategies to dampen innate immune responses [42].

Following the activation of the innate immune system, adaptive immune responses are induced. Early studies in individuals who succumbed to EVD described robust T cell activation, followed by a collapse in the T cell population [43,44]. More recent studies demonstrated the impact of T cell dynamics, kinetics and phenotype on EVD outcome [44,45,46]. It is known that T cells, particularly CD8^+^ T cells, are important in viral infection. However, the role of CD4^+^ and CD8^+^ T cells in protection is still under discussion. For example, it was demonstrated that CD8^+^ T cell deficient mice, but not CD4^+^ T cell or B cell deficient mice, succumbed to a subcutaneous infection with a mouse-adapted EBOV strain suggesting a crucial role of CD8^+^ T cells in protection [47]. Following the 2013–2016 West Africa epidemic, more studies analysed the activation of T cells in EVD survivors. We reported that the dominant CD8^+^ polyfunctional T cell phenotype was IFNγ^+^, TNF^+^, IL-2^−^ in EVD survivors in Guinea [48]. We also found that both CD4^+^ and CD8^+^ T cells contributed to specific T cell memory but with a differing cytokine profile. CD4^+^ T cells produced IFNγ, TNFα and IL-2, whereas CD8^+^ T cells only produced IFNγ and TNFα [49]. Sakabe et al. found that CD8^+^ responses to the NP were immunodominant in survivors in Sierra Leone [50]. More details about T cell responses following a natural EBOV infection can be found in several reviews [46,51]. Regarding the humoral response, some studies have suggested that early development of IgM and IgG was associated with a positive outcome [52] while antibody deficiencies were reported in fatal cases [53]. Furthermore, monoclonal antibodies (mAbs) were isolated from EVD survivors, some of which were shown to be neutralising and protective in animal models [54,55,56,57,58,59]. However, it is still debated whether the antibody titres or the neutralisation activity of antibodies is the stronger forecaster of protection [60]. This review will discuss the role of neutralising antibody responses following a natural infection and in the context of therapeutics and vaccination.

## 2. Neutralising Antibodies Following a Natural Infection

### 2.1. Neutralising Antibodies against EBOV

Mechanisms of antibody activity include neutralisation and Fc receptor-mediated effector functions (e.g., antibody-dependent cellular cytotoxicity, which is the killing of a target cell coated with antibodies by an effector immune cell; antibody-dependent cellular phagocytosis, which is a mechanism of clearance of antibody-coated pathogens or tumour cells by macrophages and natural killer cells and antibody-dependent complement deposition, which is the deposition of a complement component on infected cells mediated by IgG or IgM).

In the context of EBOV infection, it is not fully clear which function is the most associated with protection [46]. However, the characterisation of neutralising antibodies isolated from survivors and their subsequent evaluation in animal challenge models helped to improve our insight into neutralising antibody responses. It is crucial to underline that a lot of studies that analysed antibody responses to EBOV focused on IgG, especially total IgG in serum. However, the neutralisation activity can also involve pentameric IgM and monomeric/dimeric IgA present in serum and particularly abundant in mucosae [61].

Some studies suggested that neutralising antibody levels were modest early post-infection in humans but increased over time. Luczkowiak et al. analysed the neutralising activity of the plasma of three 2013–2016 West Africa epidemic survivors using a lentiviral EBOV-GP pseudotyped infection assay. They found that neutralising antibody titres increased up to 9 months post-infection [62]. Another study in Western patients confirmed this. Williamson et al. analysed B cell responses in four acute *Ebolavirus*-infected patients that had been repatriated to the US during the 2013–2016 West Africa epidemic. They found that between 1 and 3 months post-recovery, there was a low frequency of EBOV-specific B cells encoding for antibodies that displayed low neutralising activity. However, one neutralising antibody isolated in this study led to protection in a mouse EBOV challenge model [63]. A high diversity of neutralising antibodies may be needed for an efficient neutralisation, which could explain the delay in the development of neutralising responses [46]. However, it was clearly observed in several studies that long-term survivors developed robust and sustained neutralising antibody responses mainly targeting GP. Antibodies able to neutralise a pseudotyped EBOV GP were detected in survivors from the 1976 Yambuku outbreak 40 years later [64]. In addition, mAb 114, a potent neutralising mAb against pseudotyped EBOV GP lentivirus particles was first isolated from a survivor of the 1995 Kikwit outbreak, 11 years post-recovery [54]. This mAb is now a component of the recently approved drug named Ebanga^®^. The presence of persistent neutralising antibodies was confirmed in a larger cross-sectional study. Halfmann et al. measured anti-EBOV-specific humoral responses in 214 survivors and 267 close contacts (including 56 healthcare workers) in Sierra Leone, 15–32 (median 28) months post-recovery [65]. The study revealed 97.7% of survivors had antibodies against at least one of the three antigens (GP, VP40, NP) with 85% harbouring antibodies against all three. Of the survivors with a detectable antibody response against GP, all but one had neutralising titres, typically in the range of 1:128 to 1:152, but some exceeded >1:2048 [65]. A longitudinal study performed by Thom et al. also confirmed the maintenance of total and neutralising antibody responses over the course of several years. Thom et al. performed a longitudinal study looking at 117 survivors and 66 contacts, sampling patients from 3 to 14 months post-discharge, with follow up collections 12 and 24 months later [48]. The study found similar findings to Halfmann et al., where at 3–14 months 96% of survivors developed IgG specific response and this correlated well with total antibody titres (*r* 0.85; *p* < 0.0001). Interestingly, 96% of survivors maintained high titres of neutralising antibody against live Ebola virus, with a mean of 1/174. This is ten-fold greater than the titres observed in patients one-month post-vaccination with an EBOV vaccine, however, it is not known what it means for the protection as correlates of protection are unclear [48]. A smaller longitudinal study investigating B cell responses carried out by Davis et al. followed the four 2013–2016 West Africa epidemic survivors repatriated to the US (previously mentioned) from discharge to 2 years post-infection. Davis et al. concluded that IgM levels typically declined after a few months, while IgG and IgA levels remained elevated [66]. This was hypothesised to be a result of antigen restimulation. However, opposing evidence by Thom et al. indicated no antigen stimulation of T cells was detected during their longitudinal study, suggesting there might be a different reason for the retention of high titres. This topic is still debated [48]. Davis et al. also characterised EBOV GP-specific monoclonal antibodies and they found that only a subset was capable of recognising cell-surface GP, a subset that contained neutralising antibodies [66]. Some research groups characterised the regions targeted by the neutralising antibodies. Bornholdt et al. analysed 349 monoclonal antibodies specific to GP isolated from a 2014 EBOV Zaire outbreak survivor. They found that 77% of these antibodies could neutralise live EBOV. After analysing the epitopes recognised by the neutralising antibodies, they reported that mAbs which targeted the GP stalk region proximal to the viral membrane were particularly effective at protecting mice against lethal EBOV challenge [59]. Recently, Khurana et al. described specific sites on GP mounting a neutralising antibody response in rabbits, which were protective in a lethal EBOV mouse model [67]. EBOV GP regions targeted by neutralising antibodies will be discussed in the Section 5.2.

Recently, Gunn et al. described the profile of humoral responses in EVD survivors from Sierra Leone. Interestingly, this study highlighted the development of both neutralising and polyfunctional IgG1 and IgA in survivors [68]. It is probable that there is a synergy between the neutralising activity and the innate immune effector functions via Fc receptor of antibodies. To our knowledge, it has not been shown in the context of EBOV, but it has been recently described in the context of SARS-CoV-2 [69].

The delay in neutralising antibody response early post-infection but long-term persistence of neutralising antibodies in survivors may suggest a role of neutralising antibodies in the clearance of the Ebola virus rather than in the early stages of infection resolution. The characterisation of GP epitopes inducing protective neutralising antibodies, as well as a better understanding of antibody polyfunctionality at systemic and mucosal levels could be very valuable and may help to improve the efficacy of current antibody-based therapeutics and vaccines.

### 2.2. Neutralising Antibodies against Non-EBOV Filoviruses

Aside from EBOV epidemics, SUDV, BDBV and MARV outbreaks make up the vast majority of other recorded human filovirus infections and so it is important to consider the analysis of immune responses to these viruses. This is especially true because historically their small outbreaks have often occurred in remote regions of Africa with limited health infrastructure which restricts the collection of patient samples and data. Consequently, our understanding of these diseases is less detailed and, despite being equally deadly, there are currently no licensed therapeutic options for non-EBOV filoviruses.

Sobarzo et al. analysed the neutralising antibody responses of survivors of the 2000–2001 Gulu [70] and 2012 Kibaale [71] outbreaks caused by SUDV. Serum samples were collected 12 years or 3 years post-recovery, respectively. Similarly to EBOV, a robust and persistent SUDV-specific antibody response, mainly targeting GP and NP, was observed in both cohorts. Five of five Kibaale survivors and five of six Gulu survivors displayed antibodies capable of neutralising the whole SUDV by plaque reduction neutralisation assay (PRNT) [70,71].

Marburg has been responsible for two outbreaks exceeding case numbers of 100. Stonier et al. performed a longitudinal study following PCR positive survivors of the 2012 MARV outbreak in Uganda sampling patients 9, 15, 21 and 27 months after the outbreak [72]. All survivors had an antibody response to MARV GP and NP and most had a response to VP30 and VP40. Whilst total antibody titres remained high in all patients throughout the study, at 9 months, only two of six survivors demonstrated, via PRNT, a neutralising antibody response between 1:20 and 1:40 which diminished by 21 months and disappeared by month 27 [72]. These results suggest a more rapid waning of neutralising antibodies following MARV infection compared to EBOV infection. Could this difference be linked to the presence of extra GP products with EBOV, which is not observed with MARV? The mechanisms behind these results are unclear.

Other researchers isolated monoclonal antibodies from peripheral blood B cells from survivors of the 2007 BDBV outbreak in Uganda. They found a large proportion of BDBV GP-specific monoclonal antibodies including some antibodies which strongly neutralised BDBV but also SUDV and EBOV using chimeric filoviruses [57].

Together these results show that robust neutralising antibody responses are also induced following an infection with non-EBOV filoviruses.

## 3. Antibody-Based Therapeutics and Vaccines

### 3.1. Monoclonal Antibodies

Therapeutic mAbs have been a field of great interest. Two EBOV specific mAb therapeutics, REGN-EB3 (Inmazeb™, Regeneron Pharmaceuticals, Tarrytown, NY, USA) and mAb114 (Ebanga™, Ridgeback Biotherapeutics, Miami, FL, USA) having recently been approved by the Food and Drug Administration (FDA, Silver Spring, MD, USA).

Monoclonal antibodies can be created in a laboratory or isolated from a specific B cell clone from survivors or vaccinated animals. Monoclonal antibody therapy is the passive transfer of immunity in the form of one, or a small number of, antibodies targeting a single epitope of a protein, usually GP [73]. This differs from polyclonal antibodies elicited from vaccination or infection where antibodies target a higher number of epitopes on the viral protein [73]. One of the advantages is that mAbs have an immediate and potent effect and can be used as a post-exposure therapeutic and can also be administered prophylactically with a relatively long half-life, e.g., mAb114 24.2 days [56,74].

The GP_1_ head is home to the RBD and therefore an obvious target for mAbs including mAb 114. Mab114 is the only effective monotherapy and is licensed under the name Ebanga™ following impressive results from the Pamoja Tulinde Maisha (PALM) trial where it demonstrated a protective effect by reducing mortality to 35.1% [75,76]. Odesivimab is also part of the licensed Inmazeb™ cocktail but it is non-neutralising because it binds to the GP_1_ head but a little further away from the RBD than mAb114 and consequently does not neutralise (Table 1) [77]. FVM04 is a broadly neutralising antibody that binds to the GP_1_ head of the GP and can neutralise EBOV and SUDV in addition to providing good efficacy in mouse models (Table 1) [78].

The GP_1_ base is potentially the most common target for neutralising antibodies. It is the target of KZ52, which is a research standard for neutralisation, even though it was shown not to be protective in NHP trials. KZ52 binds to an epitope on the GP_1_ base that at least partially overlaps with several other potential therapeutics e.g., 2G4 and 4G7 [79]. 2G4 and 4G7 are both neutralising antibodies from the ZMapp cocktail which share overlapping epitopes on the GP_1_ base (Table 1) [80]. Maftivimab neutralising antibody of the Inmazeb™ cocktail also binds to the GP_1_ base. While it does not share an epitope with the aforementioned mAbs of the Inmazeb™ cocktail, it is thought to have the same mechanism of neutralisation (Table 1) [77].

The GC is a highly variable region that is usually thought of as non-neutralising, but it was found to be a target for several therapeutic mAbs. Saphire et al. via the Viral Haemorrhagic Fever Consortium demonstrated that antibodies targeting GC were mostly non-neutralising but a small subset of neutralising antibodies were detected [81]. The GC is also a target for Atoltivimab which is a neutralising antibody in the licensed Inmazeb™ cocktail and able to activate effector functions (Table 1) [77,81]. Furthermore, the GC contains the epitope for the broadly neutralising EBOV-548. This mAb binds to a conserved GC epitope in EBOV and BDBV. It can neutralise both viruses and effectively activates effector functions [82]. EBOV-520, which binds to the GP base region, strongly demonstrates co-operative binding. It binds to its epitope with 5× greater affinity once EBOV-548 is bound to the GC. Together, EBOV-520 and EBOV-548 can fully protect NHPs from mortality, though EBOV-520 alone has also demonstrated partial protection in animal models [82,83,84]. Finally, 13C6 is a non-neutralising antibody from the ZMapp cocktail that binds to the GC but is removed during cathepsin cleavage. It also effectively activates Fc functions [80].

ADI-15878 is a neutralising pan-ebolavirus mAb which binds to conserved regions on the HR1 and IFL domains of EBOV, SUDV and BDBV [85]. When administered alone, survival rates post-challenge in guinea pigs of 33–50% were observed. However, when paired with ADI-23774 in a cocktail called MBP134 at a high dose, it provided 100% protection [86]. CA45 is another pan-ebolavirus mAb with neutralising activity against EBOV, SUDV, BDBV and RESTV that binds to the IFL and GP_1_. This mAb was shown to be protective in animal models, both individually and in combination with FVM04 in NHP trials (Table 1) [87]. CA45 and FVM04 can also be given in a cocktail with MR191 an anti-MARV and RAVV mAb, that provides 100% protection against NHPs challenged with MARV when tested alone and within the cocktail [87,88]. MR191 binds to residues in the RBD that are essential for receptor binding and are hence highly conserved among related filoviruses. MR191 however, does not neutralise EBOV because the GC obscures the epitope whereas MARV has a disordered and flexible GC that gives the mAb access to this epitope [89,90].

The PALM trial is the only large clinical trial to date evaluating anti-EBOV experimental therapeutics in a field setting during the 2018 outbreak in the Congo, with patients presenting during various stages of disease following positive RT-PCR. It compared ZMapp, mAb114, REGN-EB3 and Remdesivir in 681 participants in a 1:1:1:1 ratio. One striking result was that ZMapp and Remdesivir were found to have no statistically significant effect on reducing EBOV mortality which stood around 50% [76]. The study did find however, that REGN-EB3 (Inmazeb™) and mAb114 (Ebanga™) had a protective therapeutic effect, reducing mortality to 33.5% and 35.1%, respectively. The effectiveness of the treatments was especially high among patients who received treatment early, with an estimated 11% increase in the risk of mortality with each day the treatment was delayed. Furthermore, patients who were classified to have a high viral titre also had a much higher mortality of 67% even with mAb114 or REGN-EB3 [76].

### 3.2. Convalescent Plasma

Convalescent plasma therapy is the passive transfer of immunity from a convalescent donor to a patient with an acute infection [95]. The transfer of convalescent blood product to a patient is an old therapy which was already used to treat various infections in humans and animal models at the end of 1800’s [96]. The theoretical basis for its use against EBOV infection is that EVD survivors are thought to be protected from re-infection through humoral immunity and so the transfer of this protective sera containing EBOV-specific antibodies could have therapeutic benefits for patients. EBOV outbreaks take place in low-income countries where expensive drugs are not always affordable and so convalescent plasma may provide a more cost-effective treatment solution. However, it comes with its complications as donor blood must be tested for a wide variety of contaminants including HIV, malaria and hepatitis A which are known to have high prevalence in many regions that suffer EBOV outbreaks [95]. Convalescent plasma therapies were tested to reduce the risks of transfusion-transmitted infections and their impacts on anti-EBOV antibodies were assessed. For example, Amotosalen/UVA pathogen reduction technology was tested to treat EVD convalescent plasma. Plasma were analysed by two types of ELISA and three neutralisation assays and it was found that anti-EBOV titres remained relatively unchanged following the treatment [97].

During the 2013–2016 West Africa epidemic, the Ebola Tx trial in collaboration with The Conakry Ebola Survivors Association organised a large plasma collection programme in Conakry in Guinea between November 2014 and July 2015 [98]. This consortium evaluated the efficacy of convalescent plasma in comparison with standardised supportive care in EVD patients in a phase 2/3 open-label non-randomised trial setting. Even though no serious adverse effects were observed with the use of the convalescent plasma, the therapy was found to have no significant effect on mortality compared to the control group (31% of patients treated with convalescent plasma died 3–16 days post-diagnosis, compared to the 38% in the control group) [99]. These results may be due to the varying levels of EBOV-specific neutralising antibody in the plasma. At that time, they did not have any test to measure EBOV-specific antibody titres and neutralising antibodies in the field. Other clinical trials evaluated some methods to screen the antibody responses in donors’ plasma in order to potentially select the plasma with high neutralising antibody titres. Brown et al. measured anti-EBOV antibodies in EVD survivors in Liberia in 2014–2015 using two ELISA and two neutralisation assays (microneutralisation, PRNT) [100]. They found that the four assays were concordant to measure donor antibody titres. However, 15 of 100 donors, including seven with a confirmed EBOV PCR positive result, did not have any detectable EBOV-specific antibodies. This trial found that viral load was reduced in EVD patients who received the convalescent plasma containing higher antibody levels, but not in patients who received the therapy with lower antibody levels [100]. Tedder et al. performed a similar evaluation in Sierra Leone using an IgG capture competitive double-antigen bridging enzyme immunoassay and a pseudotyped virus assay [101]. Both studies demonstrate the benefit of screening donors’ plasma for neutralising antibody titres even though neutralising antibodies are yet to be validated as correlates of protection.

### 3.3. Vaccine-Induced Neutralising Responses

#### 3.3.1. EBOV Vaccines

The role of antibody responses in protection following vaccination with a replication competent recombinant VSV virus encoding for Zaire EBOV Kikwit 1995 GP (rVSV-ZEBOV) was firstly analysed in preclinical studies. Marzi et al. vaccinated five NHP groups 28, 21, 14, 7 and 3 days before challenge [102]. Surprisingly, an induction of EBOV-GP specific IgG responses was already reported 3–7 days post-vaccination in NHPs. The research group found a partial protection of animals vaccinated 3 days before the challenge and a full protection of animals immunised ≥7 days before the challenge. This indicates that rVSV-ZEBOV may elicit very rapid humoral response and protection [102]. Another preclinical study reported that antibodies were sufficient at protecting mice from infection following immunisation with rVSV-ZEBOV. Meanwhile the depletion of CD8^+^ T cells did not compromise protection [103]. Similar results were observed by Marzi et al. in NHPs. CD8^+^-depletion did not impact on survival of rVSV-vaccinated animals following a challenge [104]. Wong et al. also analysed the role of humoral response following rVSV-ZEBOV vaccination in NHPs and found significantly higher total IgG titres in the serum of animals which survived post-challenge compared to the non-survivors [105]. The same team also confirmed long-term protection by challenging vaccinated animals 6–12 months post-immunisation with rVSV-ZEBOV. They observed that the levels of EBOV GP-specific IgG antibody, measured immediately before challenge, correlated with protection, whereas neutralising antibody were not always a reliable measure of protection in their animal model [106]. Another research group determined whether rVSV-ZEBOV vaccine could be used as a post-exposure treatment in Rhesus macaques. They found that four of eight macaques were protected if treated up to 30 min following a lethal infection. While the differences in cellular responses where minimal between the animals that survived and those that succumbed to the virus, there was a significant difference in neutralising responses. Indeed, neutralising antibodies were detected on days 14–36 post-challenge in animals that survived the infection, while the humoral response was not detected in animals that succumbed to the infection, suggesting a critical role for the humoral response [107]. Another study showed that NHPs vaccinated with rVSV-ZEBOV were also protected from EBOV aerosol challenge. Interestingly, upon measurement of circulating rVSV-ZEBOV specific-IgG responses post-vaccination and post-challenge, antibody responses were found to increase post-challenge. While the neutralising capacity of the antibodies was not analysed, they found there was no evidence of IFNγ or TNFα production in CD4^+^ or CD8^+^ before or after the challenge, which further suggests a major role of humoral response in protection [108]. Qiu et al., compared the protective immune responses in NHPs immunised with rVSV-ZEBOV by intramuscular (IM), intranasal (IN) or oral route (OR). They observed that IgG, IgA and IgM antibody responses were detected in the serum of all vaccinated animals independent of the route of vaccine administration. Post-challenge, IgG and IgA titres increased, while IgM titres did not exceed the levels observed post-vaccination. Globally, IgM titres ranked IN ≅ OR > IM, with an IN titre 2.4 times higher than IM, while IgA and IgG responses ranked IN > OR > IM (IN 6.8-fold > IM) and IN > OR ≅ IM (IN 9.0-fold > IM), respectively. They also analysed the level of neutralising antibodies in the sera 21 days post-vaccination and a few days prior to the challenge. Whilst neutralising antibody titres were relatively low in all animals, the OR (OR > IN >IM) produced the highest neutralising antibody titres. This study clearly highlighted the impact of immunisation with rVSV-ZEBOV on the induction of neutralising antibody responses [109]. Interestingly, NHPs previously infected with simian-human immunodeficiency virus were vaccinated with rVSV-ZEBOV. Following EBOV challenge, 4/6 animals survived. None of the six animals had a detectable antibody response by the day of challenge but three animals which survived developed a modest antibody response post-challenge, which suggests a role of antibodies in survival of these animals [110].

Recombinant VSV-ZEBOV (Erbevo^®^, Merck, Kenilworth, NJ, USA) is now the first fully licensed EBOV vaccine. This vaccine is delivered in humans in one dose and its safety and efficacy were also evaluated in clinical trials. Open-label, dose-escalation phase 1 trials were performed in healthy volunteers in Europe and Africa. They measured a persistent EBOV GP-specific antibody response in all vaccinated participants and higher neutralising responses when individuals received a higher dose of rVSV-ZEBOV vaccine [111]. Other phase 1 clinical trials reported very similar results in the US, Canada or Europe. In each study, a dose-dependent neutralising response was observed but occasionally neutralising responses failed to be detected in some individuals who received a lower dose of rVSV-ZEBOV vaccine [112,113,114]. One study provided clinical data on rVSV-ZEBOV efficacy, a phase 3 trial in Guinea, where a ring vaccination program was employed to include cases, contacts and contacts of contacts. The study reported 100% efficacy after 10 days and found that, 32 days following the detection of the first case in a vaccinated cluster, no new cases were reported, highlighting the ability of this vaccine to prevent transmission [115]. Despite the potential for bias relating to the standard of care between the experimental and control groups in the protocol of this study (as reported by some researchers), this latter highlights the high efficacy of the rVSV-ZEBOV vaccine in humans [116]. Halperin et al. showed that rVSV-ZEBOV produced strong antibody titres with 94% of participants becoming seropositive after 28 days and 91% remaining seropositive after 24 months. Geometric mean titres (GMT) rose from below the assay detection limit of 36.11 to 1262.0 after 28 days before gradually decreasing but retaining a high GMT of 920 after 24 months. The GMTs of neutralising antibodies, measured by PRNT, were high and continued to increase, peaking at 18 months with a plateau at 24 months [117]. Interestingly, Khurana et al. studied the human antibody repertoire in individuals vaccinated with rVSV-ZEBOV. They reported a high initial neutralising IgM immune response before IgG becomes the dominant subtype, which may explain the rapid protection provided by the vaccine. They demonstrated a higher diversity of antibody epitopes in vaccinees who received 20 million plaque-forming units (PFU) compared to those who received 3 or 100 million PFU. Another finding was that higher levels of neutralising GP-specific antibodies were induced after a single vaccination with 20 or 100 million PFU. A boost did not improve neutralising antibody response [118].

Adenoviral vector vaccines have been also developed in the context of EBOV. These vaccines are well known to induce robust T cell responses [119,120,121]. Wong et al. evaluated the antibody responses in the context of vaccine candidates based on AdHu5 expressing Zaire EBOV GP [105]. They compared the immune responses and survival rates of knockout mice (Rag-1^−/−^, B cell^−/−^, CD8^+^ T cell^−/−^, IFN-γ^−/−^, and CD4^+^ T cell^−/−^), reporting that B cell and CD4^+^ T cell responses were the most critical in the development of a protective immune response against a mouse adapted strain of EBOV [105]. When repeated in guinea pigs, the average titres of anti-EBOV GP antibodies and neutralising antibody post-challenge were significantly higher in survivors than in non-survivors [105]. Interestingly, they also performed a similar study in NHPs. In this animal model, they found a higher anti-GP IgG response in survivors, but they did not detect any differences in neutralising antibody response between the survivors and non-survivors. However, a difference in T cell responses between the cohorts was detected in this latter model [105]. Chen et al. observed a durable EBOV-neutralising response in mice vaccinated with a prime-boost vaccine regimen based on a chimpanzee serotype 7 adenovirus expressing EBOV GP and a truncated version of EBOV GP1 protein [122].

In humans, the two-phase vaccine candidate Ad26.ZEBOV (Zabdeno^®^) and MVA-BN-Filo (Mvabea^®^), produced by Johnson&Johnson (New Brunswick, NJ, USA), is a vaccine candidate that has progressed well through clinical trials and has received marketing authorisation by the European Commission in July 2020. This is a viral vector vaccine, that uses two heterologous doses. The first dose is a non-replicating Adenovirus type-26 encoding EBOV GP and the second dose is a multivalent recombinant Modified Vaccinia Ankara vector-based vaccine encoding the GPs from EBOV, SUDV, MARV and the nucleoprotein of TAFV [123]. Anywaine et al. compared the immunogenicity of heterologous two-dose Ad26.ZEBOV and MVA-BN-Filo vaccination regimens in a phase 1 trial in Uganda and Tanzania. They demonstrated a robust immunogenicity when Ad26.ZEBOV was administered as the first dose followed by MVA-BN-Filo. This regimen promoted higher neutralising and total antibody titres with 93% of participants achieving seropositivity at the time of the second dose, reaching 100% 21 days after the second dose. Neutralising antibody titres were initially low but peaked at 21 days following the second dose 100% of participants in the 56-day interval group, before decreasing and stabilising 180 days post-initial vaccination. Titres plateaued until the final time-point at day 365 [124]. A similar phase 1 clinical trial was run in Kenya, where high levels of neutralising GP-specific antibodies were detected and sustained up to 360 days after the first dose [125].

Globally, preclinical and clinical studies demonstrated a key role of antibody responses in protection post-vaccination, particularly in the context of rVSV-ZEBOV. However, the correlation between neutralising responses and protection following vaccination is still debated.

Finally, there is uncertainty regarding the efficacy of postexposure mAb treatments following a recent vaccination with an EBOV vaccine. This scenario of exposure may happen early post-injection when vaccination is not yet fully effective. Cross et al. demonstrated in rhesus macaques that vaccination with rVSV-ZEBOV 1 day prior to EBOV challenge followed by anti-EBOV GP mAb MIL77 treatment 3 days later increased the rate of survival compared to animals vaccinated or treated with MIL77 only [126]. However, additional data is needed to draw a robust conclusion about a potential synergy between vaccination and antibody-based immunotherapy.

#### 3.3.2. MARV Vaccine Candidates

To date no MARV vaccine has been approved yet, but several vaccine platforms have been attempted, including a recombinant VSV vector expressing MARV GP which showed efficacy in NHPs both post-exposure and prophylactically [127]. Jones et al. demonstrated that a single intramuscular injection led to 100% protection of vaccinated NHPs from a lethal MARV challenge [128]. Daddario-DiCaprio et al. confirmed these results with the same vaccine by proving the 100% protection following a challenge with heterologous MARV strains and RAVV [129]. Both studies reported high IgG titres in vaccinated NHPs but an absent or very low neutralising response. More recently, Mire et al. conducted a study involving six NHPs immunised with a rVSV-MARV-GP platform prior to challenge 13 months later. Immunisation elicited strong total antibody titres (between 1600 and 12,800) which remained elevated throughout the study. Regarding the neutralising antibody responses determined by PRNT, all NHPs had a neutralising response at day 28 post-vaccination but the titres were relatively low and by the day of challenge, two had no neutralising antibodies [130]. Together, these results suggest a less important role of neutralising antibodies in protection from MARV challenge.

Interestingly, a follow-up study by Daddario-DiCaprio et al. showed that immunisation with rVSV-MARV vaccine up to 20–30 min post-challenge protected all NHPs. All treated animals showed low to moderate amounts of IgM by day 6 post-challenge. Four of the five vaccinated animals developed a moderate IgG response by day 10 post-challenge. However, PRNT demonstrated low amounts of neutralising antibodies between days 6 and 37 in the plasma of all immunised animals [131]. Similar results were confirmed by Geisbert et al. In addition, a partial protection was reached when the vaccine was administered 24 h and 48 h post-challenge, when five out of six and two out of six macaques were protected, respectively. All animals which survived to challenge showed moderate to high levels of MARV-specific IgG in sera, whereas animals that died did not have any detectable IgG responses [132]. The sum of these results suggests the importance of humoral response following rVSV-MARV vaccination even though the neutralising activity may be less important with rVSV-MARV vaccine compared to rVSV-EBOV vaccine.

## 4. Methods to Measure Neutralising Antibody Responses

Researchers use different assays to evaluate neutralising antibody titres in serum or other biological fluids. To measure EBOV-specific neutralising antibody responses, the gold standard is the PRNT, which uses authentic live virus to measure the number of plaques formed upon infection of a cell line, such that if the sample is capable of neutralisation, fewer plaques would be observed [133]. Briefly the biological fluid (e.g., serum) containing the antibodies is mixed with the authentic live virus, usually for 1–2 h, and used to infect a permissive cell line. An agarose overlay is added for 5–7 days to avoid a ‘too rapid’ spreading of the virus. Finally, the cells can be fixed and plaques are counted, often manually. The PRNT takes several days and its development at a large-scale can be limited. It is the reason why microneutralisation assays (MNA) were also developed in the context of EBOV [97]. The concept of MNA is very similar to PRNT but 1 h post-incubation of cells with the mix EBOV/serum, the cells are washed and fixed for 1–2 days. Finally, infected cells can be detected using anti-EBOV and secondary antibodies. The number of spots can be counted using an imaging system [58]. Both methods, PRNT and MNA, are robust. However, EBOV is classified as a biosafety safety level (BSL) 4 organism as it is a very dangerous pathogen. BSL-4 work requires Class III safety cabinets, very specific pressure conditions in the lab, very strict decontamination procedures of the materials and trained staff. This increases the time and cost of such experiments, as well as limiting the number of sites with such capabilities [134].

Pseudotyped virus neutralisation assays can be a more flexible and manageable approach. Pseudotyped viruses (pseudoviruses) are recombinant viruses with core and envelope proteins derived from different viruses. They carry full or partial/modified sequence genomes to give rise to replication proficient or deficient virus. A reporter gene (e.g., luciferase, green fluorescent protein) is incorporated into the genome of the vector to detect a reduction of luminescence or fluorescence by neutralising antibodies, following infection of a permissive cell line. In the context of EBOV, recombinant VSV or lentiviral viruses are often used as a backbone to express EBOV GP. The advantage is that the recombinant pseudotyped viruses can usually be used at lower containment levels than EBOV. However, data generated from different pseudovirus systems can vary. Indeed, parameters such as the type of backbone, the reporter system and the expression of GP on the backbone may impact the sensitivity and the specificity of the assays, as well as their ability to accurately detect neutralising antibodies. Therefore, some studies have investigated conditions to improve the correlation with live EBOV neutralisation assays. We compared side by side two systems of EBOV GP pseudoviruses. Steeds et al. showed that the VSV luciferase pseudovirus system had a greater correlation (r = 0.85 + *p* < 0.0001) to the live EBOV assay than the HIV-1 system (r = 0.54 + *p* = 0.0004) [135,136]. Similarly, Wilkinson et al. ran a study across several laboratories and reported that labs using the VSV system reported higher correlation with a wildtype Ebola virus neutralisation assay (r = 0.84 and r = 0.96). Konduru et al. also confirmed a high correlation between the VSV platform and live EBOV neutralisation assay [137,138]. A limitation of the pseudovirus system is the absence of proteins aside from GP. While neutralisation antibodies mainly target EBOV GP, we cannot exclude the possibility that neutralising antibodies may be directed against other key proteins (e.g., VP40, other forms of GP) and cannot be detected using a pseudovirus system. For instance, VSV systems do not produce sGP, which could have an impact on neutralisation outcomes. Interestingly, Saphire et al. evaluated the impact of sGP on the results generated by three different neutralisation assays: VSV platform, authentic live EBOV, EBOVΔVP30-RenLUc virus. The VSV system did not express sGP whereas the other systems expressed wildtype sGP. Globally, they observed that the presence of sGP did not prevent neutralisation and in some cases, neutralisation was higher in the sGP-expressing assays [81]. When pseudoviruses are replication-deficient, this system can only detect the antibodies which prevent the entry of the virus into the cell. For example, the system cannot detect the impact of antibodies on the production of new viral progeny.

Pseudovirus platforms are surrogate systems to identify neutralising antibodies. However, the production of recombinant filoviruses expressing a reporter protein, which can help to monitor and quantify the infection, may be useful to improve the characterisation of antibodies able to neutralise EBOV or other filoviruses. Thus, an alternative approach to measuring filovirus-specific neutralising antibody responses is to use chimeric filoviruses. Chimeric filoviruses are man-made viruses composed of components from two filoviruses. Usually, one live filovirus is used as a backbone and its GP is replaced by a heterologous filovirus GP (e.g., EBOV GP). The potential difference in neutralisation-sensitivity between a chimeric filovirus and a pseudovirus has to be considered in order to avoid producing misleading results in the detection of neutralising antibodies [139]. Llinykh et al. compared the neutralisation-sensitivity of VSV-expressing filovirus GP with EBOV-expressing heterologous filovirus GP from BDBV, SUDV, MARV and LLOV. They reported that chimeric filoviruses were as sensitive as authentic filoviruses expressing the same GP. However, VSV chimeric viruses were more sensitive to antibody neutralisation than authentic filoviruses [139]. The analysis of neutralising antibody responses can also be performed by surrogate virus neutralisation tests. Whilst these tests have been broadly developed for other viruses such as SARS-CoV-2, to our knowledge they have yet to be developed for EBOV or other filoviruses.

To conclude, the choice of platform used to measure neutralising antibody responses is crucial in order to yield the most accurate results. However, the neutralising antibody titres generated with different platforms may be difficult to compare. It is the reason why the use of a WHO research standard in each assay can help to compare the titres between the studies and consequently, the neutralisation efficacy determined in each study.

## 5. Viral Neutralisation

### 5.1. Main Mechanisms of Neutralisation

Neutralising antibodies bind epitopes which block critical amino acids in the virus’s lifecycle, typically epitopes involved in cell entry or the production of new viral progeny. The different mechanisms of neutralisation are shown in Figure 2 (pink boxes).

The vast majority of EBOV neutralising antibodies are targeted towards the GP as it contains multiple crucial epitopes exposed on the surface, particularly in the RBD [140]. Neutralising antibodies which bind to the EBOV GP block the interactions between the EBOV GP and NPC1 and prevent receptor binding via a direct binding to the RBD or in a way to block access to the RBD, thus preventing any subsequent binding [90]. The presence of neutralising antibodies can also prevent EBOV fusion with the endosomal membrane, usually necessary for the virus to escape into the cytoplasm. Indeed, neutralising antibodies can impede this by binding to the GP_1_/GP_2_ interface, clamping the GP and preventing the conformational changes in the GP required for this process (Figure 2) [17]. Finally, cathepsin cleavage usually removes the GC and MLD to expose the RBD mediating access for receptor binding. Neutralising antibodies can bind to GP in such a way to partially or totally prevent cleavage. For example, mAb CA45 disrupts cleavage to produce partially cleaved GPs, while mAb100 almost completely inhibits cleavage [54,140,141]. Though not clearly described in the context of EBOV, some antibodies can neutralise by altering the structure or conformation of the viral GP to the point that it is unable to perform a critical function, e.g., receptor binding (see Figure 2) [142]. VP40 is critical for viral budding and the production of new progeny and therefore its potential as a therapeutic target has been investigated. For example, anti-VP40 mAbs have been shown to neutralise EBOV by preventing the formation of new progeny (Figure 2). This is bolstered when administrated in cocktails with mAbs targeting different sites on the VP40 protein or with GP-specific antibodies [143,144]. Antibodies, particularly polymeric IgM and IgA, can also agglutinate virus particles, forming aggregates that reduce the contact between the virus and the cell and affect subsequent entry, in addition to signalling, for antibody-dependent phagocytosis (Figure 2) [145,146,147].

Finally, it is important to highlight that based on some studies performed by the Viral Haemorragic Fever Immunotherapeutic Consortium, 5% of non- or weakly neutralising EBOV-specific antibodies tested in the panel were found to be protective when assessed in vivo proving a crucial role for non-neutralising antibodies in protection [81]. Perhaps more protective antibodies have not yet been discovered, as the assays of choice for the analysis of antibody responses are largely based on neutralisation only. Indeed, non-neutralising antibodies can stimulate an array of processes that aid in viral clearance including initiating complement deposition, contributing to viral inactivation, phagocytosis and complement-dependent cytotoxicity (Figure 2) [148]. Monoclonal antibodies are also capable of interacting with Fc receptors activating antibody-dependent cell-mediated cytotoxicity and antibody-dependent cellular phagocytosis [68,146,148,149]. Furthermore, it has been reported that non-neutralising antibodies have been shown to have synergistic neutralising ability, whereby they bind to the viral GP, altering the structure of an epitope making it more accessible for neutralising antibodies to bind. This has been documented for both EBOV and SUDV [150]. The three main mechanisms of non-neutralising antibodies are mentioned in Figure 2 (yellow boxes). Some monoclonal antibodies isolated from EBOV survivors have been shown, at sub-neutralising concentrations, to cause antibody-dependent enhancement via antibody-dependent phagocytosis, especially in macrophages which are thought to be the preferred target cell of EBOV. This could play a potential role in antibody-dependent enhancement of EBOV infection leading to a more severe disease and worse clinical outcomes [151].

### 5.2. Regions Targeted by Neutralising Antibodies

Antibodies binding to the GP head are frequently neutralising because they block the function of the RBD, hence inhibiting cell entry and GP binding to NPC1 which is required for escaping the endosome.

The Viral Haemorrhagic Fever Immunotherapeutic Consortium investigated 168 EBOV specific mAbs for their ability to neutralise live virus and pseudotyped VSV and measured their ability to protect BALB/c mice following a challenge with mouse-adapted EBOV. The study found that antibodies specific for the GP_1_ head, base, the HR2 and IFL provided statistically significant protection. All of the same domains, except the GP_1_ base were targeted by more potently neutralising antibodies. Consequently, the neutralising ability of the antibodies showed a statistically significant correlation with protection, with a Spearman’s rank value of 0.65 (the closer rank value is to 1, the stronger the association between the ranks), based on the percentage of infected cells using an authentic EBOV [81]. HR1, HR2 and IFL are also important neutralisation targets given that HR1 and HR2 both facilitate IFL insertion into the host membrane [12]. Some studies clearly described that the neutralising activity of CA45 or ADI-15878 and ADI-15742 were linked to IFL binding [58,93]. Finally, the base of GP_1_ is also a target for neutralisation, when an antibody binds, it prevents the release of the base which clamps HR1 and HR2 to instigate the conformational changes required for viral escape from the endosome, thereby preventing these conformational changes from taking place [12].

The GC and MLD are designed to shield GP from humoral responses and are cleaved by cathepsin in the endosome. They play no part in the receptor binding or fusion of the EBOV GP with the cell membrane, hence are not always targets for neutralisation [81]. This is demonstrated by the potently neutralising mAb114 and the non-neutralising 13C6 which have overlapping epitopes binding to the GP_1_ core and at least partially to the GC. MAb114’s binding to the GC isn’t critical and remains bound and neutralising when the GC is removed following cathepsin cleavage. Meanwhile, residues on GC are critical for 13C6 binding and when removed, 13C6 no longer remains bound to the EBOV GP, rendering it non-neutralising and allowing EBOV to freely interact with NPC1 [75].

## 6. Strategies to Prevent Immune Escape from Neutralising Antibodies

The efficacy of antibody-based therapeutics and vaccine-induced antibody responses may be reduced over time due to viral immune evasion. Immune evasion may impair antibody binding and consequently neutralisation reducing protection provided by the mAbs-based therapeutics, vaccines or a previous natural response.

A major drawback of mAbs is that they are specific to one epitope. Consequently, they are more vulnerable to immune escape from rapidly mutating viruses, and this has been observed with many other viruses such as influenza, HIV or SARS-CoV-2 [77,152,153,154,155]. In an experimental setting, Steeds et al. showed that GP variants G74R, P330S and H407Y were capable of readily escaping the WHO research neutralisation standard KZ52. Worryingly, KZ52 has overlapping epitopes with several other promising mAb therapeutics, e.g., CA45, 4G7 and more (Table 1) [135]. A solution to this concern is the use of mAb cocktails, that target several distinct epitopes, hence reducing the chance of immune evasion. It is the reason why numerous mAbs discussed previously are part of a cocktail. Furthermore, this strategy also allows the addition of non-neutralising antibodies which may be more effective at initiating effector functions to increase the efficacy of the drug e.g., 13C6 or Odesivimab [77,80]. Another tactic that is employed involves targeting highly conserved epitopes that are often critical to viral fitness.

Currently, there are only two licensed EBOV-specific mAbs, with no licensed therapeutics on the market for the treatment of any other filovirus. Thus, pan-ebolavirus or pan-filovirus mAbs would significantly improve filovirus treatment options. An example of which includes CA45 and FVM04 that when used in combination as a cocktail provide 100% protection against both EBOV and SUDV in NHP and mouse models [87,93]. Furthermore, when MR191, is added to the cocktail, it also provides 100% protection against MARV [87]. Interestingly, some pan-neutralising mAbs were isolated from a 2013–2016 West Africa EBOV epidemic survivor and showed protection in mouse and ferret models [58]. Gilchuk et al. also isolated mAbs from the 2013–2016 West African epidemic and 2018 DRC outbreak survivors targeting epitopes at the base region of the GP which displayed pan-neutralising and protective abilities in mice, guinea pigs and ferrets [83]. Some survivors from the 2014 Boende EVD outbreak also mounted pan-filovirus serum neutralising responses [156].

There has also recently been a push for cross-reactive pan-filovirus vaccine candidates. Experimental DNA- and VSV-based vaccines encoding multiple GPs of various filoviruses have been tested in NHP models. For example, Keck et al. immunised two cynomolgus macaques three times with a trivalent GP cocktail consisting of EBOV/SUDV/MARV. Sera from the macaques had strong IgG measured by ELISA. While neutralising titres against the GP of all three filoviruses, were detected, although significantly lower neutralisation was observed with MARV. As MARV is more distantly related, it is not surprising that less cross-reactivity is shared [157].

The development of pan-neutralising therapeutics and vaccines is likely the future of filovirus research. These therapies may prove to be a highly beneficial resource in the poor regions where filoviruses are endemic.

## 7. Conclusions

This review has summarised that antibodies, and in particular neutralising antibodies, are an important line of defence against filovirus infection but current research falls short of defining them as a correlate of protection. Preclinical and clinical studies were not able to determine the level of neutralising antibodies necessary to protect an individual.

Monoclonal antibodies are the first licensed post-exposure therapeutics for EVD, which, coupled with expanded use of vaccines, have the potential to save a large number of lives. However, vaccine access in the field is still limited and mAb therapies are not perfect. In addition, the emergence of EBOV variants which could reduce the efficacy of mAbs and vaccine-induced humoral responses has to be considered. Consequently, newer and more effective antibody-based treatments containing more broadly neutralising antibodies, as well as the development of vaccines based on epitopes eliciting cross-reactive neutralising antibodies, would be very beneficial.

## Figures and Tables

**Figure 1 pathogens-10-01201-f001:**
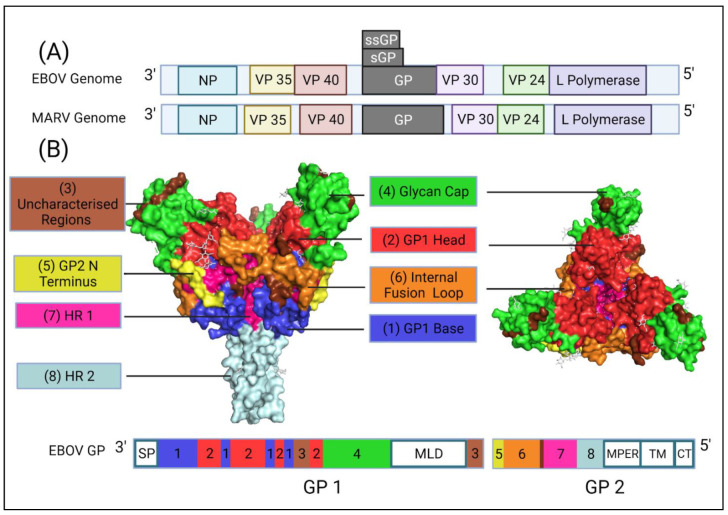
(**A**) A comparison of the EBOV and MARV genome structures highlighting key differences such as how EBOV has multiple GP gene products and where there are differences in overlapping genes. (**B**) A diagram of the EBOV GP with domains highlighted based on the work of Lee et al. 2009 [12]. Domains of the EBOV GP as they appear on the gene: GP_1_, SP (signal peptide), GP_1_ base residues: (33–69, 95–104, 158–167 and 175–189), GP_1_ Head: (70–94, 105–157, 168–175 and 214–226), Glycan Cap: (227–310), MLD Mucin-like-Domain, and in GP_2_, N-terminus of GP_2_, Internal Fusion Loop: (511–553), HR1 (554–598), HR2 (599–630), MPER Membrane proximal external region, TM transmembrane domain, CT Cytoplasmic tail. Regions highlighted in brown are uncharacterised as they could not be assigned to a domain due to differences with the protein structure used and the domains as listed by Lee et al. 2009 (32, 191–195, 210–211, 470–478) and the protein structure used from Zhao et al. [12,13]. Regions of the EBOV GP in white boxes with blue outlines are regions that could not be shown via X-ray diffraction and so do not appear on the GP structure in the diagram, in addition to residues (28–30, 196–209, 284–285, 294–300, 431–469, 632–669) [12,13]. Diagram was created with BioRender.com (©BioRender 2021, accessed in June 2021) and the protein structure was generated in The PyMOL Molecular Graphics System, Version 2.0 Schrödinger, Delano Scientific LLC, Berkeley, CA, USA using the PDB accession: 5JQ3.

**Figure 2 pathogens-10-01201-f002:**
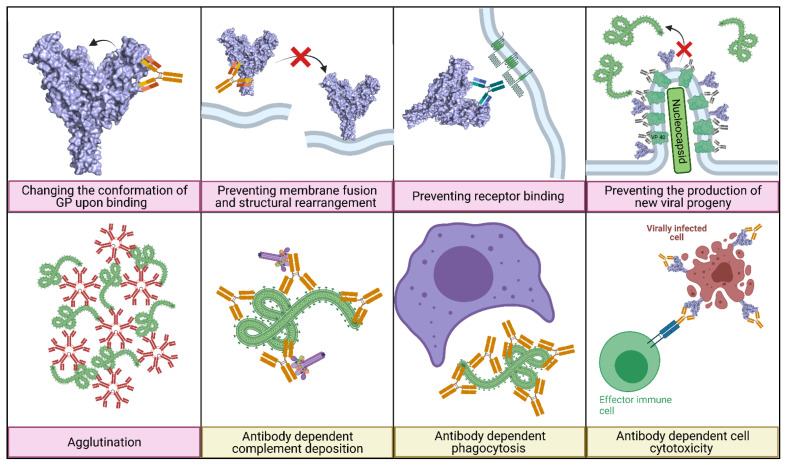
The mechanisms of action of neutralising antibodies (pink boxes) and non-neutralising antibodies (yellow boxes). Diagram was created with BioRender.com (©BioRender 2021, accessed in June 2021) and the protein structure was generated in The PyMOL Molecular Graphics System, Version 2.0 Schrödinger, Delano Scientific LLC, Berkeley, CA, USA using the PDB accession: 5JQ3.

**Table 1 pathogens-10-01201-t001:** A table describing the characteristics and epitopes of several key mAbs, with a figure integrated highlighting the residues on the EBOV GP that make up the epitopes of each of the mAbs in the table. The critical/known residues are in red, and the putative residues highlighted in yellow. The protein structure was generated in The PyMOL Molecular Graphics System, Version 2.0 Schrödinger, Delano Scientific LLC, Berkeley, CA, USA using the PDB accession: 5JQ3.

Antibody (Cocktail)	Epitope	Brief Description—How It Was Discovered and Where It Targets	Neutralising
**KZ52—WHO Research Standard** Critical residues—red: 511, 550, 552, 552, 556 Putative residues—yellow: 24, 40, 43, 507-508, 513-514, 549, 551	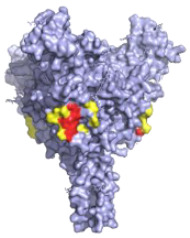	A potently neutralising antibody isolated from a survivor of the 1995 Kikwit outbreak binding to the GP_1_/GP_2_ interface to prevent insertion of the fusion loop into the membrane [80].	✓
**13C6 (ZMapp)** Critical residues—red: 270, 272	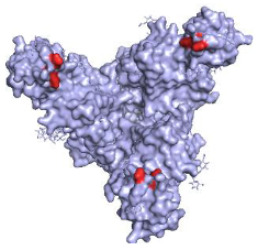	A non-neutralising humanised mouse antibody that binds to the GP_1_ head and GC regions and can activate effector functions [80].	X
**2G4 (ZMapp)** Critical residues—red: 511, 550, 553, 556	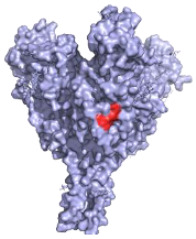	A neutralising humanised mouse antibody that binds to the GP_1_/GP_2_ interface to prevent insertion of the fusion loop into the endosome membrane [80].	✓
**4G7 (ZMapp)** Critical residues—red: 511, 552, 556	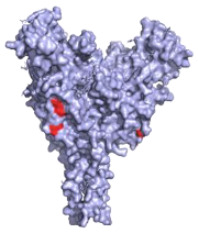	A neutralising humanised mouse antibody that binds to the GP_1_/GP_2_ interface to prevent insertion of the fusion loop into the membrane [80].	✓
**Ansuvimab-zykl/mAb114 (Ebanga^™^)** Known residues—red: 111-119	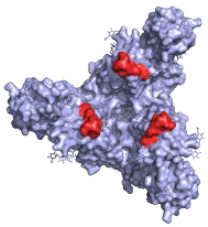	Isolated from a survivor of the 1995 Kikwit outbreak that targets the RBD blocking access to NCP1 [74,91].	✓
**Atoltivimab/REGN3470 (Inmazeb^™^)** Putative residues—yellow: 236-244, 264-297 and 298-308	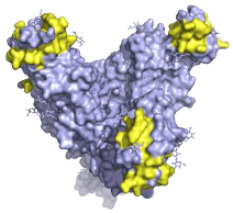	Developed by the VelcoImmune platform in which genetically engineered mice express fully human antibodies. Atoltivimab is a neutralising antibody that binds to GC, it is also able to activate effector functions [92].	✓
**Odesivimab/REGN3471 (Inmazeb^™^)** Putative residues—yellow: 114 to 122, 139 to 151, 236 to 244, and 265 to 287	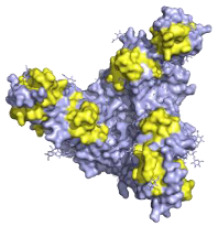	A non-neutralising VelcoImmune antibody that binds to the GP_1_ head but further from the residues involved in receptor binding compared to mAb114. It is very capable of activating effector functions [92].	X
**Maftivimab/REGN3479 (Inmazeb^™^)** Putative residues—yellow: 531 to 545	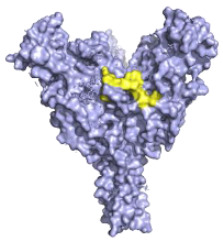	A neutralising VelcoImmune antibody that binds to the GP_1_/GP_2_ interface to prevent insertion of the IFL into the endosome membrane [92].	✓
**CA45—Pan-ebolavirus** Critical residues—red: 64, 517, 546, 550 Putative residues—yellow: 38, 40-41, 66, 68, 101-104, 184, 186-187, 211-213, 513-516, 518-519, 544-545, 547-549, 551-552, 554, 558	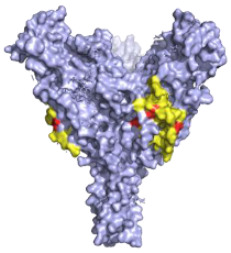	Isolated from challenged NHPs, CA45 binds to the IFL and GP_1_/GP_2_ interface preventing insertion of the IFL into the endosome membrane and hence neutralises EBOV, SUDV, BDBV and RESTV. CA45 provides protection in mice, guinea pigs and ferrets. Also gives 100% protection to NHPs when given in a cocktail with FVM04 and MR191 [87,93,94].	✓
**FVM04—Pan-ebolavirus** Known residues—red: 115, 117-118	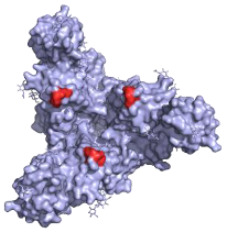	Targets the RBD blocking interactions with NPC1. It neutralises EBOV, SUDV and BDBV, though only protective from EBOV and SUDV challenge in mouse and Guinea pig model. Can be given in cocktail with CA45 and MR191 [78,87].	✓

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
