# Peer review of "Filovirus Neutralising Antibodies: Mechanisms of Action and Therapeutic Application"

_pathogens, 2021, doi:10.3390/pathogens10091201_

Round 1

Reviewer 1 Report

Overall, a nice review that covers filovirus phylogeny, Ebola virus entry, the Ebola virus glycoprotein, immunity in humans and animal models, detailed discussion of mechanisms of neutralization, longitudinal response in humans, monoclonal antibody therapeutics, convalescent plasma, vaccine, and touches on immune escape. Minor revisions, clarifications, and additional citations would improve the manuscript.

  1. The figures are very informative yet are not referred to in the text.
  2. Heading of 1.2. Suggest breaking up into three sections:

Lines 60-75 could stay as “Genome Organisation of Ebolavirus”

Lines 76-135 could be titled something like “Cellular entry of Ebola”

Lines 136-156 could be titled something like “EVD and immunity”

Line 128: should be “micropinocytosis”

Lines 150-152 feels out of place; suggest moving it to the paragraph below where T cells is discussed.

  1. Line 168. Please include additional citations as there is an extensive body of work from a number of different investigators across the globe who have developed and tested monoclonal antibodies against Ebola virus beyond the one that is cited.
  2. Line 185-186. As it is currently written, the text indicates that Regeneron’s VelcoImmune mice are being used to screen therapeutics for efficacy against Ebola virus. The VelcoImmune mice have been engineered to express human immunoglobulin variable domains with mouse constant regions and are used to generate monoclonal antibody therapeutic candidates rather than screen them for protection against Ebola. More typically, the antibody candidates isolated from VelcoImmune mice are then tested in other animal models (guinea pig, NHPs). Please revise the sentence.
  3. One of the main challenges in studying development of natural adaptive immunity in animal models is that virtually every animal model for Ebola without intervention is a lethal model, with animals often dying before an antibody response develops. I think the authors are trying to make this point in lines 193-198, but it gets lost a bit in the discussion of dosing. It may be beneficial here to make the distinction between animal models that are designed to test vaccine and therapeutic efficacy vs models that are aimed at dissecting Ebola pathogenesis and pathology.
  4. Line 302. As r values are included for the VSV-luciferase and HIV-1 pseudovirus, it would be great to have the r value for the VSV-GFP system included if it is available.
  5. Line 311. Please elaborate on the section describing replication competent chimeric viruses – chimeric with what?
  6. The sentences within lines 306-308 appear to contradict each other as currently written.
  7. Lines 365-368. Recent work including Saphire et al 2018 have shown that while most glycan cap-directed antibodies do not have neutralizing activity, a subset of antibodies display high levels of neutralizing activity, and Murin et al (10.1016/j.celrep.2021.108984) show that an epitope within the glycan cap is highly conserved and may be a site for broadly neutralizing activity.
  8. There have been several neutralizing antibodies described that target the internal fusion loop beyond the ones that the authors cite (10.1016/j.cell.2017.04.037; 10.1016/j.cell.2017.04.038
  9. Suggest including “vaccines” in the section title for “antibody-based therapeutics”.
  10. Line 448 capitalize MARV
  11. Line 502-507. It would be useful to discuss whether antibodies are correlates of protection for the vaccines and include a discussion of whether antibodies were induced in the SHIV-infected NHP study.
  12. Marzi et al (10.1126/science.aab3920) showed rapid protection by the rVSV vaccine in NHPs that would be useful to discuss in this section – mechanism is very likely antibody-independent given the time frame, but as the authors propose the use of monoclonal antibodies with vaccination, it would be informative to discuss the potential mechanisms of synergy between mAb-based therapeutics and vaccines
  13. Lines 475-477 appear to be mixed/incomplete sentence.
  14. Recent efforts to identify broadly neutralizing antibodies across ebolaviruses or filoviruses often target conserved epitopes that are essential for viral fitness. These antibodies (alone or in a cocktail) might be worthwhile discussing in the immune escape section as a strategy to prevent immune escape.
  15. Include references for HIV and influenza immune escape (line 549)
  16. Suggest rewording the sentence in line 563-567

Author Response

Overall, a nice review that covers filovirus phylogeny, Ebola virus entry, the Ebola virus glycoprotein, immunity in humans and animal models, detailed discussion of mechanisms of neutralization, longitudinal response in humans, monoclonal antibody therapeutics, convalescent plasma, vaccine, and touches on immune escape. Minor revisions, clarifications, and additional citations would improve the manuscript.

The authors thank the reviewer for the nice comments. The requested modifications can be found below.

  1. The figures are very informative yet are not referred to in the text.

All figures have been referred to in the text now.

  1. Heading of 1.2. Suggest breaking up into three sections:

Lines 60-75 could stay as “Genome Organisation of Ebolavirus”

Lines 76-135 could be titled something like “Cellular entry of Ebola”

Lines 136-156 could be titled something like “EVD and immunity”

As suggested, the text of the first section has been broken up into the suggested headings (Lines 64, 80, 142).

Line 128: should be “micropinocytosis”

We wrote micropinocytosis so we presume the reviewer wanted to say “it should be macropinocytosis”. The word has been modified (Line 133).

Lines 150-152 feels out of place; suggest moving it to the paragraph below where T cells is discussed.

The paragraph about T cells has been fully reorganised and expanded. Some references have been added. This section has been rephrased to improve the clarity (Lines 159-175).

  1. Line 168. Please include additional citations as there is an extensive body of work from a number of different investigators across the globe who have developed and tested monoclonal antibodies against Ebola virus beyond the one that is cited.

Indeed, a lot of research groups have worked on the development and testing of antibody-based therapeutics in animal models and clinical trials. We decided to merge the section “2.1.1. Animal models” and “2.1.2 Longitudinal immune responses in humans” into a section “Neutralising antibodies against EBOV” (Line 186-270) and developed other sections “Monoclonal antibodies” (Lines 305-376), “Convalescent plasma” (Lines 396-437) and “Vaccine-induced neutralising antibody responses” (Lines 440-598). As suggested by the reviewer, we integrated much more references and we also tailored the messages for each section.

  1. Line 185-186. As it is currently written, the text indicates that Regeneron’s VelcoImmune mice are being used to screen therapeutics for efficacy against Ebola virus. The VelcoImmune mice have been engineered to express human immunoglobulin variable domains with mouse constant regions and are used to generate monoclonal antibody therapeutic candidates rather than screen them for protection against Ebola. More typically, the antibody candidates isolated from VelcoImmune mice are then tested in other animal models (guinea pig, NHPs). Please revise the sentence.

In our first manuscript, there was a little bit a mix between the analysis of antibody responses following a natural infection and the evaluation of monoclonal antibodies. The section about the analysis of neutralising antibody responses following EBOV infection (Lines 186-270) and the section about monoclonal antibodies (Lines 305-376) have been fully reorganised.

  1. One of the main challenges in studying development of natural adaptive immunity in animal models is that virtually every animal model for Ebola without intervention is a lethal model, with animals often dying before an antibody response develops. I think the authors are trying to make this point in lines 193-198, but it gets lost a bit in the discussion of dosing. It may be beneficial here to make the distinction between animal models that are designed to test vaccine and therapeutic efficacy vs models that are aimed at dissecting Ebola pathogenesis and pathology.

The role of animal models in EBOV research, their development as well as their limitations have been fully reviewed in Longet et al. 2020 (10.3389/fimmu.2020.599568). Thus, we decided to refocus the general message of the review about the analysis of neutralising responses observed in humans post-infection, post antibody-based therapies or post-vaccination and in the text we integrated the animal studies as a tool to determine the protection. However, we decided not to mention the development of animal models in this review.

  1. Line 302. As r values are included for the VSV-luciferase and HIV-1 pseudovirus, it would be great to have the r value for the VSV-GFP system included if it is available.

The r number for the Wilkinson study has been added and the sentence has been rephrased (Lines 635-638).

  1. Line 311. Please elaborate on the section describing replication competent chimeric viruses – chimeric with what?

As suggested by the reviewer, the paragraph about chimeric viruses has been developed and clarified (Lines 652-662).

  1. The sentences within lines 306-308 appear to contradict each other as currently written.

The section containing this sentence has been now clarified (Lines 552-665).

  1. Lines 365-368. Recent work including Saphire et al 2018 have shown that while most glycan cap-directed antibodies do not have neutralizing activity, a subset of antibodies display high levels of neutralizing activity, and Murin et al (10.1016/j.celrep.2021.108984) show that an epitope within the glycan cap is highly conserved and may be a site for broadly neutralizing activity.

Both studies have been now mentioned to support the statement that neutralising GC-specific antibodies are less frequent but some are able to bind a conserved epitope leading to broadly neutralising activity (Lines 336-339 and 341-344).

  1. There have been several neutralizing antibodies described that target the internal fusion loop beyond the ones that the authors cite (10.1016/j.cell.2017.04.037; 10.1016/j.cell.2017.04.038

Both studies have been included in the text (Line 351-355, 740-743).

  1. Suggest including “vaccines” in the section title for “antibody-based therapeutics”.

Thanks for this suggestion. In the new manuscript, we have a section named “Antibody-based-therapeutics and vaccines” (Line 304) including three sub-sections “Monoclonal antibodies”, “Convalescent plasma” and “Vaccine-induced neutralising responses”.

  1. Line 448 capitalize MARV

The sentence has been modified and all capitalised abbreviations have been checked (Line 795)

  1. Line 502-507. It would be useful to discuss whether antibodies are correlates of protection for the vaccines and include a discussion of whether antibodies were induced in the SHIV-infected NHP study.

Based on the new structure of the review, we clarified its focus. Along the manuscript, we discussed the studies and mentioned if they suggested antibodies as correlates of protection.

Giesbert et al. 2008 (10.1371/journal.ppat.1000225) has been included in the section dedicated to vaccines (Lines 485-490). Antibody responses were not induced in SHIV-NHP post-vaccination with rVSV-ZEBOV by the day of challenge but modest antibody responses were detected in ¾ animals which survived post-challenge.

  1. Marzi et al (10.1126/science.aab3920) showed rapid protection by the rVSV vaccine in NHPs that would be useful to discuss in this section – mechanism is very likely antibody-independent given the time frame, but as the authors propose the use of monoclonal antibodies with vaccination, it would be informative to discuss the potential mechanisms of synergy between mAb-based therapeutics and vaccines

We added the study Marzi et al. in the manuscript (Lines 443-448). And as suggested by the reviewer, we discussed the potential synergy between mAb-based therapeutics and vaccines in the section dedicated to vaccines (Lines 514-521).

  1. Lines 475-477 appear to be mixed/incomplete sentence.

This specific sentence was unclear and has been deleted.

  1. Recent efforts to identify broadly neutralizing antibodies across ebolaviruses or filoviruses often target conserved epitopes that are essential for viral fitness. These antibodies (alone or in a cocktail) might be worthwhile discussing in the immune escape section as a strategy to prevent immune escape.

Indeed, the development of broadly neutralising antibodies is a strategy to prevent the immune escape. Consequently, we decided to merge the section dedicated to cross-reactive antibodies and the section dedicated to immune escape into a new section named “Strategies to prevent Immune Escape from neutralising antibodies”. The whole section has been reorganised. Additional studies have been described (Lines 777-788).

  1. Include references for HIV and influenza immune escape (line 549)

As suggested, the references have been added (Line 764).

  1. Suggest rewording the sentence in line 563-567

The conclusion has been fully reworded (Lines 802-813).

Reviewer 2 Report

This review paper describes the role of neutralizing antibodies in filovirus infection and disease, both those naturally produced and those that have been exploited in therapeutics, some of which have been approved for use in humans. The authors refresh readers on filovirus biology, the mechanisms by which filoviruses enter the host cell, as well as how that process is thwarted by antibodies, particularly neutralizing antibodies. They further describe the current state of the art in filovirus antibody research methodologies and detail several of the antibody-based therapies currently being developed or approved.

The paper is well researched, reasonably well written and is a well-focused summary of a specific type of antiviral therapy. The paper is digestible and easily accessible even to non-filovirus experts, and smartly avoids much of the complicated and jargon-rich immunological word soup that often make these kinds of papers difficult to follow or inaccessible to those who aren’t fluent in immunobiology.

That said, there are ways in which the paper could be improved. While there do not appear to be any specific “dealbreaker” issues that would necessarily impede publication, the authors’ could find several of these suggestions valuable and strongly worth considering in order to make their work clearer and more informative to their target audience, particularly 1) the inclusion of MARV to the subject matter rather than a sole focus on EBOV, 2) the addition of background information on typical filovirus disease course and concurrent immune responses in the context of antibody development, production and protection, 3) more consistent detail about each of the antibodies presented including logistics, dosing regimen, efficacy, etc., and 4) any through-line message the authors wish to convey to readers, if any, such as that relating to correlates of protection.

Below are the main general comments that typically apply across the paper, presented in no particular order. This is followed by minor line-by-line comments that are generally editorial in nature.

  1. The abstract seems to put weight on topics that are not fully explored in the review itself. While correlates of protection are mentioned, it does not come across as a major recurring theme as the abstract suggests. The abstract suggests that the authors will go through studies that define nAbs as correlates of protection, only for them to later state that there are no correlates of protection for EBOV, even nAbs that seem to have protective effects. Thus the mention of CoP in the abstract feels almost contradictory. Further, there is limited discussion of evasion by filoviruses in the paper beyond a small section, while the abstract gives the impression it would have more weight. Finally, while it is refreshing on one hand to read a paper on antibody responses that can avoid the complexities of immune cell types, cell markers and immune pathways, the abstract does imply there would be elements of humoral and cellular adaptive responses, but they are largely avoided in the actual text.
  2. While the relative lack of immunology-heavy jargon is refreshing, it would be beneficial to readers, especially those less familiar with filoviruses, to include a broad summary of what is known about the humoral and adaptive response in NHPs and humans as it relates to antibody production/protection during a typical filo infection, including clinical timing, cell types and tissues involved, antibody titers, what impact titers have, if any, on protection, etc. In other words, walk readers through the usual series of events. It would make most sense to blend this with the information already present on page 4 (paragraphs 2-4), made into a dedicated separate section ideally following viral genome organization and cell entry, especially as these paragraphs currently seem out of place being tacked onto the end of this section.
  3. There seems to be a lack of opportunity to discuss MARV in this review, as it is just as vital a subject, just as potentially deadly as EBOV, has caused almost as many known outbreaks as EBOV (including the one just announced in Guinea) and is possibly even more important from a public health standpoint as drug research is more limited and there are fewer therapeutics developed. Further, MARV already has its genome shown in Fig 1 and is mentioned prominently in section 2.1.3, so it might as well be given similar weight to EBOV in the context of this review.
  4. What is the ultimate message of this review, if any? While reviews are indeed a summary of the literature, they often attempt to use that literature to draw conclusions, highlight gaps in knowledge or conduct meta-analysis that can provide additional insight to the field. This is not strictly needed in a review, but if the authors do wish to focus their readers’ attention to any key themes, suggestions, or take-home messages regarding neutralizing antibodies to filoviruses, this should be made clear and be used to tie together the information discussed. This helps give the review a distinctive “voice” and a connective thread beyond simply being “study A showed this, study B showed that”, etc. As written, there does not seem to be a strong overarching cohesive thread beyond the basic subject matter itself. If the lack of nAbs being correlates of protection is supposed to be the takeaway message, then the authors need to make this much more “front and center” in the paper, doing a better job of emphasizing CoP, how and/or why in each instance, abs or nAbs fail to achieve this distinction, why this matters, and possibly how this differs from other viruses that do have clear correlates of protection (potentially even, in those cases, antibodies).
  5. Make sure to check all formatting is proper and consistent throughout the paper (italics, numbers below ten written out, writing it as “, e.g. ” every time, proper formatting of viral nomenclature like “ebolavirus,” spacing between words and citation numbers, citation formatting, double spacing, improper spacing for proteins such as GP1 and GP2 like in the Fig 1 legend, unnecessary capitalization and/or hyphenation such as “Mucin-like-Domain” in Fig 1 legend, antibody dilution formatting, etc.). Also, carefully examine the use of commas throughout the paper, as there is often incorrect and/or excessive comma or semicolon use that can make sentences unclear; many examples are highlighted in the line by line comments.
  6. In the intro, it might be worth mentioning BOMV again when discussing LLOV, given both viruses were first detected in bats, as well as it being the newest known ebolavirus and with uncertain pathogenicity; mention ab data for BOMV if such data exists.
  7. HUJV and XILV could be cut from the paper as they have no impact on the review and are never revisited.
  8. Check over all figures to make sure that every box is physically aligned properly; for example, Figs 1 and 2 has misshapen, misaligned boxes. This will make the diagrams look cleaner.
  9. The authors highlight that MARV has key differences in genome structure to EBOV. As adding more information about MARV to the paper would be very beneficial, it would be good to detail what those differences are and where they lie. Perhaps discuss why MARV doesn’t seem to need the extra GP products seen with EBOV (or alternatively, why EBOV does seem to need them), how this affects the antibody response during infection, and what it ultimately means for vaccine development between the two genera.
  10. Make sure to describe or define all immune markers, cell types or processes mentioned throughout the paper (such as “antibody-dependent cell-mediated cytotoxicity”, “antibody-dependent enhancement”, etc.).
  11. Make sure to clarify what antibody dilution/titer numbers actually mean in terms of their vaccination efficiency or neutralization efficacy.
  12. The paper could use more logistical information about each of the Abs mentioned, perhaps as a separate section, including: cost, temperature, regimen requirements, when they can be administered, how they are administered, manufacture and transport, etc.
  13. In keeping with the above comment, a very useful figure would be to compare all discussed therapies in terms of % protection, timing within context of infection and/or symptomology, regimen, immune response, logistics, current licensure/approvals or pre-clinical/clinical trials progress, and potential drawbacks including those related to logistics. One figure that puts every Ab discussed from the paper together into one handy visual guide. Perhaps most easily, this could simply be a more detailed expansion of Fig 3.

Line-by-line comments:

  1. Line 36-83: change to “and Bombali”; should be “the first of these six species”, not four; define GP upon first mention on line 49; spell as “spillover” on line 51; might be more visually useful to present the filovirus proteins in genome order on lines 62-63; spell as “subunits” on line 80; change to “and it is only” on line 83.
  2. Lines 87-90: this is an incomplete and confusing sentence; “, and).” Line 93 is also an incomplete sentence.
  3. Line 94-97: remove “the” before “mucin-like domain”; add comma after “The GP1 head” and “trimeric conformation”; remove comma after “chalice”; add period after “(RBD)”; capitalize “The GP1 base” and decapitalize “heptad”; change to “heptad repeat 1”. Further, this entire sentence is generally confusingly written, and it might be best to rewrite this more clearly.
  4. Line 100: Change to “MLD and GC are highly variable” for clarity.
  5. Line 102: Is there missing info here? The sentence is incomplete.
  6. Line 105: This seems to be repetitive info compared to what was previously written on lines 95-97.
  7. In Fig 1B, the regions of GP labeled need to be better connected to the residue numbers listed with the Fig 1 legend, as it’s currently not always clear which region is which. In the legend, it might also be cleaner to put residues in parentheses for each region, rather than list them after colons.
  8. Line 124-125: incomplete sentence, revise.
  9. It might be beneficial to readers to include a simple figure that visualizes the cell entry process.
  10. Change all uses to “Type I IFN”.
  11. Line 156: clarify that CD3 is a pan T cell marker.
  12. Lines 160-162: rewrite the beginning of this paragraph, as it is generally considered improper and awkward writing to explicitly define something cited from a reference.
  13. Line 173: capitalize Animal Rule.
  14. Lines 174-178: awkwardly written and incomplete sentence, revise; remove comma after “several animal models”.
  15. Line 183: briefly define Biosafety Level 4, what the conditions are, that this is the level required for work with authentic filovirus, and why studies are costly and difficult to handle in these conditions.
  16. Line 186: detail what the VelcoImmune suite is a little more. How do studies with mice benefit the field if they don’t fully recapitulate human disease?
  17. Line 193-212: change “However” to “Meanwhile”; move “typically” to before “all NHPs succumb”; clarify that “control groups” means unvaccinated, infected NHPs; define “PFU” on line 196; add comma after “Halfmann et al.” on line 208; move “GP, VP40 and NP” to line 211 when first mentioning the three antigens on line 212.
  18. All three studies in section 2.1.2 should be together in their own paragraph as they are thematically related, and a new paragraph is meant to denote a change in subject.
  19. Lines 215-216: It’s confusing why this opening sentence is used to introduce the Thom study when Halfmann et al is also a big longitudinal study with antibody responses seen over several years (and so, given comment 31 above, this line should be moved, altered or cut). Also, don’t use “several” here as there’s only the description of the one (Thom) study, unless it’s clarified that Halfmann and Thom are representative examples of several studies that end up with similar findings.
  20. Line 222: explain why it matters that the titers are tenfold greater here.
  21. Lines 225-228: remove comma after “David et al.”; change “remaining” to “remained”; refine sentence as it is a bit confusing and there are too many uses of the root word “elevate”.
  22. Lines 229-234: remove “is” from “This was hypothesized”; move “of T cells” to after “antigen restimulation”; define antigen restimulation and why this could matter in the context of these two studies; remove comma after “however”; move comma after “including” to after “considered”; add hyphen in “highly-variable”; define what the small sample size was.
  23. Section 2.1.2 and 2.1.3 are overall a bit limited in detail and breadth, and it is unclear what conclusions you hope readers to draw from the studies described, if any. In the last paragraph of 2.1.2, briefly explain how knowledge of viral persistence and longevity of immune responses assists with each of the three topics listed, with examples of how this knowledge helps.
  24. Line 243: considering the intro specified that the focus of the review would be solely on EBOV, there is a sizeable amount of non-EBOV filoviruses discussed. It might be worth adjusting the intro to be more inclusive to other filoviruses, particularly MARV, as suggested in comments 3 and 9 above.
  25. Line 245-264: possibly change to “majority of other recorded human filovirus infections”; change the semicolon to a comma on line 247; shouldn’t it say 12 years prior on line 252? Revise to “detected in five out of six Gulu survivors” on line 254; delete “despite primary exposure being 11 years ago” as this is redundant what’s written above it in line 252; change to a period after “100 cases” on line 257; change to “sampling patients 9, 15, 21, and 27 months” on line 259; change to “and this diminished by 21 months” on line 263-264.
  26. Line 267; again, how do these different antibody titers relate to one another, especially when displayed using different metrics/tests?
  27. Line 271: possibly change “in which case” to “which suggests that”.
  28. Lines 271-272: does this mean total ab response is more important than neutralizing response? Why might it have lesser importance in protection and what does this mean about nAbs as CoP? Is there any data that relates this difference between MARV and EBOV to MARV’s lack of additional GP products? Finally, would it be useful to briefly discuss studies in the bat reservoir and any antibody responses described for that host, and how that might relate to protection (or not)?
  29. The paragraph starting on line 274 might make more sense appearing at the beginning of section 2.1.3, rather than at the end.
  30. Line 277-294: what kind of settings do they occur? Change to “settings that have limited”; underline not needed on line 284; seems like it should be “capable of neutralisation, fewer plaques” on line 289; remove the comma after “experiments” and move it to after “organism” on line 290; briefly define how BSL2 relates to BSL4 on line 293; change to “throughput” on line 294.
  31. Lines 295-315: move comma from “such as” to after “variables”; add “that” before “the VSV luciferase”; remove “2017” from line 301 and “a” from “has very high correlation to the PRNT”; add a comma after “However” in line 306; paragraph on line 309 should be part of the paragraph above it and not a new one, and it would make more sense to start the sentence as “Another limitation”.
  32. Lines 321-332: this paragraph goes through mechanisms of neutralization, but it mentions them out of order with when that process occurs during cell entry; it would be clearer and flow better to reorganize this in the order of how the events naturally proceed; on line 325, change to “Further, EBOV fuses”; remove the comma after “however” in line 326; fix the formatting on line 327 for GP2.
  33. Line 341: add “and” before “this is bolstered when”.
  34. Line 343: introduce IgM and IgA antibodies a little better and explain a little more detail about them.
  35. For Fig 2, the images should be better linked to the text to help illustrate each type of mechanism; within the figure, labeling cellular and viral factors involved in each diagram would be nice to see and aide in clarity; in the legend, clarify the sentence “as well as some methods…” which is currently confusing.
  36. Lines 354-359: not sure if this needs to say “assays” here; change to “measured”; in line 356, remove the period after “IFL” and add “that” before “provided statistically”; add period after “protection” and start a new sentence with “All of the same”; briefly explain what this Spearman’s rank value means in this context; lines 356-359 seem redundant with repeated info in line 356 vs line 358-359.
  37. Line 369: the paragraph beginning here should be a part of the previous paragraph as it is a continuation of the same train of thought; add commas after “neutralisation” and after “binds” on line 371.
  38. Explain briefly why “antibody-dependent enhancement” on line 392 might not be a good thing in the context of EBOV infection.
  39. Line 399: it might be useful to give generic ab names for Inmazeb and Ebanga in parentheses here.
  40. Line 401: no need for first use of “specific” here.
  41. Line 407: might be better to stick with use of “Ebanga” here rather than switching to “mAb114”; this would also help prevent those numbers from being right up against “24.2 days”.
  42. Lines 409-410: add comma after “necessary”; pluralize “vaccine”; briefly mention in text what those vaccines are, who makes them, etc., as it’s not clear and readers haven’t gotten to Fig 3 yet.
  43. Is there anything that can be done about the massive white space on page 10 and/or the two pages that Fig 3 takes up (which seems like a waste of space given the relatively little text used within the figure).
  44. In Fig 3, clarify that “(cocktail)” means that the name of the larger multi-Ab combination will be in those parentheses; for “Critical residues” clarify that these are GP residues; it might be clearer if the authors included arrows to the regions of GP the residues correspond to as a visual aide to the description; starting with Odesivimab down, why are there no residues of any kind listed, especially since regions of importance are still indicated in the description box? If they’re not known, indicate as such. On page 12, in the description box change to “binds to the Glycan Cap” and add a semicolon after “Cap”. Add a comma after “NHPs” in the CA45 description box; might be worth mentioning if this has been used clinically or not. In FVM04 description box, add period after “NPC1”, delete “hence is” and capitalize “Neutralises”; add comma after “BDBV” and change to “protective”. MR191 should be defined; why is this mAb not described on its own in the Fig?
  45. Line 415-416: clarify again that these are GP residues.
  46. Lines 423-432: briefly define the PALM trial, what “large” means, where/when the trial took place, etc.; remove comma after “however” on line 427; explain what this post exposure effect was, i.e. when were these patients presenting? Was it soon after exposure, or when symptomatic? If the latter, how sick in general were they? How much lower exactly than ZMapp and Remdesivir? How effective exactly for patients treated early? Give absolute numbers, not just relative ones. Remove comma after “mortality” on line 431.
  47. Line 435-441: name the drugs again for clarity; revise to “EBOV-specific mAbs, with no licensed”; add a comma after “mutate” on line 439; briefly define the concept of immune escape on line 440, as well as what KZ52 is within the text; doesn’t it also neutralize RESTV?
  48. Lines 444-458: remove the comma after “FVM04”; change to “MARV” on line 448; change to “driver” on line 450 and remove “However” from beginning of sentence; start a new sentence after “ELISA” on line 454; add a comma after “filoviruses” on line 455; on line 456, was flow cytometry used to identify B cells were GP-specific? Is the 0.06% of B cell cross-reactivity enough to protect?
  49. In section 5.2, describe how well these multi-Ab cocktails protects (particularly the one with MR191), when they’re given, the regimen, any potential downsides, and address why, if they work very well on multiple filoviruses, they are not being used clinically, or haven’t been licensed or approved.
  50. Line 468-469: add “be” before “administrated”; it might be cleaner to begin a new sentence after “patients” and start the new sentence with “Contaminants include”; add comma after “hepatitis A”.
  51. Line 475: what are some of the parameters involved?
  52. Lines 476-480: this section is confusing; rewrite for better clarity; add a comma after “protection” in line 479; what are some hypothetical examples of this “much more varied set” of correlates?
  53. Line 483-499: you might consider removing “viral vector” here as the description is too cumbersome, and more “vector” after “VSV virus” on line 484. Go into more detail on the Guinea trials, including when they occurred, who was involved, etc. Remove the comma after “only” on line 489, change “but” to “and” on line 495, move “however” to the beginning of the sentence on line 496. What do you mean by “the follow-up of vaccines”, is this supposed to be “vaccinees”? What was the actual experimental bias the authors refer to?
  54. Line 505: rephrase as “post-vaccination, though two out of six NHPs died post-challenge, suggesting” for better clarity.
  55. In section 5.4, how efficacious is Zabdeno and Mvabea? Is it protective, and if so, by how much? If this is known information, add these details to the text.
  56. Lines 508-514: clarify that Zabdeno and Mvabea is a two-phase vaccine candidate, fix spelling of Johnson&Johnson, add commas after “Mvabea and “Johnson” on line 508, add a comma after “vaccine” on line 513, change “and therefore, has the” to “with the”; on line 514, pluralize “glycoprotein”, remove commas after “from” and “SUDV” and add “and” before MARV and change “of TAFV” to “from TAFV”.
  57. Lines 515-516: this is somewhat redundant information to that in lines 510-512. Thus, the authors could rephrase this line as “The order and timing regimen used for these two vaccines was determined by immunogenicity data from phase II trials.”
  58. Line 518: when is 93% seropositivity achieved?
  59. Lines 518-522 are a bit confusing and should be rewritten for better clarity; why was the trial halted?
  60. Line 527: move “with Erbevo” after “vaccination” so it doesn’t sound like the challenge itself was done with Erbevo.
  61. Line 529: what is MIL-77? Why is this not listed in Fig 3? This should be described more.
  62. Lines 541-549: add commas after “antibodies” on line 540, after “mAb” on line 542, after “escape” on line 543, and after “Worryingly” on line 546; change “and instead” to “which instead” on line 542; remove comma after “P330S” and add “and”; clarify that KZ52 is a mAb and move the description of this standard to the section before it when KZ52 was first mentioned; in line 546, what are the overlapping epitopes and what are some examples of the other mAbs? Fix “observed” on line 549.
  63. Lines 555-566: add comma after “antibodies”, add “and” before “in particular” and remove comma after that, add comma after “infection” on line 556, add comma after “EVD” on line 560, add comma after “outbreak” on line 562, pluralize “measure” on line 563, add comma after “limited” on line 564, depluralize “mAbs” on line 565, consider changing “treatments” on line 565/566 to “therapeutics” to avoid redundancy of using the same word twice in the sentence.
  64. Line 556: why does the current research fall short exactly? Explain further.
  65. Line 563: is “program” here referring to PALM? If so, clarify.
  66. Line 566: the mention of “better access” seems to emphasize logistics as an important element in antibody treatment, which it is, but this concept of logistics and drug availability is not well emphasized throughout the rest of the paper. This could be corrected through addressing comments 12 and 13 above.

Author Response

  1. The abstract seems to put weight on topics that are not fully explored in the review itself. While correlates of protection are mentioned, it does not come across as a major recurring theme as the abstract suggests. The abstract suggests that the authors will go through studies that define nAbs as correlates of protection, only for them to later state that there are no correlates of protection for EBOV, even nAbs that seem to have protective effects. Thus the mention of CoP in the abstract feels almost contradictory. Further, there is limited discussion of evasion by filoviruses in the paper beyond a small section, while the abstract gives the impression it would have more weight. Finally, while it is refreshing on one hand to read a paper on antibody responses that can avoid the complexities of immune cell types, cell markers and immune pathways, the abstract does imply there would be elements of humoral and cellular adaptive responses, but they are largely avoided in the actual text.

First the authors would like to acknowledge the reviewer for their time and numerous suggestions which went a long way to improve the review. We have included many more studies to discuss the role of neutralising antibodies following EBOV infection and vaccination. The structure of manuscript and the focus of the message have been improved. Consequently, the abstract has been restructured. The role of neutralising antibodies was shown to be important in protection but their role as a correlate of protection has not been confirmed.

  1. While the relative lack of immunology-heavy jargon is refreshing, it would be beneficial to readers, especially those less familiar with filoviruses, to include a broad summary of what is known about the humoral and adaptive response in NHPs and humans as it relates to antibody production/protection during a typical filo infection, including clinical timing, cell types and tissues involved, antibody titers, what impact titers have, if any, on protection, etc. In other words, walk readers through the usual series of events. It would make most sense to blend this with the information already present on page 4 (paragraphs 2-4), made into a dedicated separate section ideally following viral genome organization and cell entry, especially as these paragraphs currently seem out of place being tacked onto the end of this section.

As suggested, more details on the immune response (humoral, cellular and innate responses) following EBOV infection in animal models and humans has been added in the section 1.3.

  1. There seems to be a lack of opportunity to discuss MARV in this review, as it is just as vital a subject, just as potentially deadly as EBOV, has caused almost as many known outbreaks as EBOV (including the one just announced in Guinea) and is possibly even more important from a public health standpoint as drug research is more limited and there are fewer therapeutics developed. Further, MARV already has its genome shown in Fig 1 and is mentioned prominently in section 2.1.3, so it might as well be given similar weight to EBOV in the context of this review.

We have included studies and considerations on MARV. We have discussed some anti-MARV vaccine candidates and mAbs against MARV (lines 292-297, 358-363, 557-598).

  1. What is the ultimate message of this review, if any? While reviews are indeed a summary of the literature, they often attempt to use that literature to draw conclusions, highlight gaps in knowledge or conduct meta-analysis that can provide additional insight to the field. This is not strictly needed in a review, but if the authors do wish to focus their readers’ attention to any key themes, suggestions, or take-home messages regarding neutralizing antibodies to filoviruses, this should be made clear and be used to tie together the information discussed. This helps give the review a distinctive “voice” and a connective thread beyond simply being “study A showed this, study B showed that”, etc. As written, there does not seem to be a strong overarching cohesive thread beyond the basic subject matter itself. If the lack of nAbs being correlates of protection is supposed to be the takeaway message, then the authors need to make this much more “front and center” in the paper, doing a better job of emphasizing CoP, how and/or why in each instance, abs or nAbs fail to achieve this distinction, why this matters, and possibly how this differs from other viruses that do have clear correlates of protection (potentially even, in those cases, antibodies).

Indeed, our take-home messages were not clear enough. In the revised manuscript, the message reviewing antibodies as a potential correlate of protection has been clarified throughout the review.

  1. Make sure to check all formatting is proper and consistent throughout the paper (italics, numbers below ten written out, writing it as “, e.g. ” every time, proper formatting of viral nomenclature like “ebolavirus,” spacing between words and citation numbers, citation formatting, double spacing, improper spacing for proteins such as GP1 and GP2 like in the Fig 1 legend, unnecessary capitalization and/or hyphenation such as “Mucin-like-Domain” in Fig 1 legend, antibody dilution formatting, etc.). Also, carefully examine the use of commas throughout the paper, as there is often incorrect and/or excessive comma or semicolon use that can make sentences unclear; many examples are highlighted in the line by line comments.

The authors have been through the script to standardise and correct all formatting.

  1. In the intro, it might be worth mentioning BOMV again when discussing LLOV, given both viruses were first detected in bats, as well as it being the newest known ebolavirus and with uncertain pathogenicity; mention ab data for BOMV if such data exists.

It was a good idea. We have added information about BOMV in the revised manuscript (lines 51-57).

  1. HUJV and XILV could be cut from the paper as they have no impact on the review and are never revisited.

The authors reduced the sections describing HUJV and XILV but decided to leave general information about HUJV and XILV for completeness of the phylogeny.

  1. Check over all figures to make sure that every box is physically aligned properly; for example, Figs 1 and 2 has misshapen, misaligned boxes. This will make the diagrams look cleaner.

Boxes in figures have been aligned.

  1. The authors highlight that MARV has key differences in genome structure to EBOV. As adding more information about MARV to the paper would be very beneficial, it would be good to detail what those differences are and where they lie. Perhaps discuss why MARV doesn’t seem to need the extra GP products seen with EBOV (or alternatively, why EBOV does seem to need them), how this affects the antibody response during infection, and what it ultimately means for vaccine development between the two genera.

More information about MARV has been added to the review particularly in the new section 3.3.2 MARV vaccine candidates (lines 557-598). For the lack of extra GP products in the context of MARV and its impact on antibody responses, we think it is a good point to explore. As to our knowledge, there are not specific studies which analysed this difference. We mentioned this point as an open question (lines 292-297). 

  1. Make sure to describe or define all immune markers, cell types or processes mentioned throughout the paper (such as “antibody-dependent cell-mediated cytotoxicity”, “antibody-dependent enhancement”, etc.).

These concepts have been defined (lines 187-193).

  1. Make sure to clarify what antibody dilution/titer numbers actually mean in terms of their vaccination efficiency or neutralization efficacy.

Neutralising titres give an idea about the neutralisation efficacy in a specific study. However, as correlates of protection are unclear, it is not known which specific titres may be necessary to lead to protection following vaccination. We wrote a few lines about the difficulties to compare the results of neutralisation efficacy between the studies (lines 666-670).

  1. The paper could use more logistical information about each of the Abs mentioned, perhaps as a separate section, including: cost, temperature, regimen requirements, when they can be administered, how they are administered, manufacture and transport, etc.

Even though it is a very interesting suggestion, it is not the purpose of this paper. In addition, it would be difficult based on how many of the drugs are very experimental.

  1. In keeping with the above comment, a very useful figure would be to compare all discussed therapies in terms of % protection, timing within context of infection and/or symptomology, regimen, immune response, logistics, current licensure/approvals or pre-clinical/clinical trials progress, and potential drawbacks including those related to logistics. One figure that puts every Ab discussed from the paper together into one handy visual guide. Perhaps most easily, this could simply be a more detailed expansion of Fig 3.

Whilst this is very critical information and interesting topic, this would be more appropriate for a review evaluating mAbs and their clinical use and efficacy. Whilst we certainly touch on these topics, the focus of the review is about how neutralising antibodies provide a protective effect and mAbs are just a very good example of this. We have attempted to clarify this message throughout the review.

 Line-by-line comments:

  1. Line 36-83: change to “and Bombali”; should be “the first of these six species”, not four; define GP upon first mention on line 49; spell as “spillover” on line 51; might be more visually useful to present the filovirus proteins in genome order on lines 62-63; spell as “subunits” on line 80; change to “and it is only” on line 83.

  1. A) In the revised version, the sentence has been modified (line 40)
  2. B) The authors are stating that the first 4 of the 6 species are pathogenic though this has been clarified (line 41)
  3. C) GP has been defined (line 54)
  4. D) It has been corrected (line 56)
  5. E) It has been re-ordered (line 66-67)
  6. F) It has been corrected (line 101)
  7. G) The sentence has been reworded (line 96)

  1. Lines 87-90: this is an incomplete and confusing sentence; “, and).” Line 93 is also an incomplete sentence.

Both sentences have been reworded (lines 96-97)            

  1. Line 94-97: remove “the” before “mucin-like domain”; add comma after “The GP1 head” and “trimeric conformation”; remove comma after “chalice”; add period after “(RBD)”; capitalize “The GP1 base” and decapitalize “heptad”; change to “heptad repeat 1”. Further, this entire sentence is generally confusingly written, and it might be best to rewrite this more clearly.

All of the grammatical suggestions have been included in the manuscript (line 101-106).

  1. Line 100: Change to “MLD and GC are highly variable” for clarity.

The suggestion has been added (line 108).

  1. Line 102: Is there missing info here? The sentence is incomplete.

The paragraph has been reworded (lines 111-115).

  1. Line 105: This seems to be repetitive info compared to what was previously written on lines 95-97.

This section 1.2 of the manuscript has been extensively modified, we hope the repetition has been reduced (lines 111-115).

  1. In Fig 1B, the regions of GP labelled need to be better connected to the residue numbers listed with the Fig 1 legend, as it’s currently not always clear which region is which. In the legend, it might also be cleaner to put residues in parentheses for each region, rather than list them after colons.

Parentheses have been added to clarify the residue numbers in the figure legend (line 122-128). A colour code has been used to identify the regions. However, the identification of each residues on the picture would impact on the clarity of the figure. The authors decided not to add more arrows or information on the picture itself.

  1. Line 124-125: incomplete sentence, revise.

The sentence has been clarified by adding more detail (‘better access to the RBD’) (line 135).

  1. It might be beneficial to readers to include a simple figure that visualizes the cell entry process.

Whilst it might be relevant, the authors considered the Figure 3 dedicated to mechanisms of neutralising antibodies was adequate to understand the concepts and that a new figure would not add a lot of value to this review. 

  1. Change all uses to “Type I IFN”.

When we mentioned Type I Interferon, we wrote Type I IFN. (For example, line 153.)

  1. Line 156: clarify that CD3 is a pan T cell marker.

This study mentioning CD3 has been removed in the revised manuscript.

  1. Lines 160-162: rewrite the beginning of this paragraph, as it is generally considered improper and awkward writing to explicitly define something cited from a reference.

In line with comments from other reviewers, this section has been fully modified.

  1. Line 173: capitalize Animal Rule.

Same comment as above. The discussion of the animal rule has been removed as it was not the focus of this review.

  1. Lines 174-178: awkwardly written and incomplete sentence, revise; remove comma after “several animal models”.

The section has been removed along with the rest of the discussion on animal models.

  1. Line 183: briefly define Biosafety Level 4, what the conditions are, that this is the level required for work with authentic filovirus, and why studies are costly and difficult to handle in these conditions.

The authors clarified these points (lines 614-618).

  1. Line 186: detail what the VelcoImmune suite is a little more. How do studies with mice benefit the field if they don’t fully recapitulate human disease?

This section has been removed along with the rest of the discussion on animal models.

  1. Line 193-212: change “However” to “Meanwhile”; move “typically” to before “all NHPs succumb”; clarify that “control groups” means unvaccinated, infected NHPs; define “PFU” on line 196; add comma after “Halfmann et al.” on line 208; move “GP, VP40 and NP” to line 211 when first mentioning the three antigens on line 212.

The authors would like to take the time to thank the reviewer for their very detailed comments throughout the review which have definitely elevated the manuscript greatly, but as stated previously this section has been removed.

  1. All three studies in section 2.1.2 should be together in their own paragraph as they are thematically related, and a new paragraph is meant to denote a change in subject.

The paragraphing structure has been changed accordingly and more studies have been added to the manuscript (line 202-256).

  1. Lines 215-216: It’s confusing why this opening sentence is used to introduce the Thom study when Halfmann et al is also a big longitudinal study with antibody responses seen over several years (and so, given comment 31 above, this line should be moved, altered or cut). Also, don’t use “several” here as there’s only the description of the one (Thom) study, unless it’s clarified that Halfmann and Thom are representative examples of several studies that end up with similar findings.

Halfmann et al. is a cross-sectional study with samples taken just once 2 years post-infection whereas Thom et al. sampled the same cohort on a yearly basis (line 228-229). The paragraphs have been restructured appropriately (line 202-256) and the word ‘several’ has been removed (line 227).

  1. Line 222: explain why it matters that the titers are tenfold greater here.

This has been clarified in the text (lines 235-237).

  1. Lines 225-228: remove comma after “David et al.”; change “remaining” to “remained”; refine sentence as it is a bit confusing and there are too many uses of the root word “elevate”.

The section has been extensively edited (line 238-247).

  1. Lines 229-234: remove “is” from “This was hypothesized”; move “of T cells” to after “antigen restimulation”; define antigen restimulation and why this could matter in the context of these two studies; remove comma after “however”; move comma after “including” to after “considered”; add hyphen in “highly-variable”; define what the small sample size was.

  1. A) Sentences have been reworded and many of the grammatical changes adopted (Line 241-245).
  2. B) Antigen restimulation and its potential impact are still debated. We have clarified this point in the manuscript (Line 241-245).
  3. C) This section has been extensively modified. Consequently, these suggestions fell into the paragraphs that were cut.

  1. Section 2.1.2 and 2.1.3 are overall a bit limited in detail and breadth, and it is unclear what conclusions you hope readers to draw from the studies described, if any. In the last paragraph of 2.1.2, briefly explain how knowledge of viral persistence and longevity of immune responses assists with each of the three topics listed, with examples of how this knowledge helps.

More studies have been described in these sections now (line 186-303).

The conclusions relating to the role of neutralising antibodies have been highlighted (line 264-270).

We mentioned antigen restimulation and its debate (lines 241-245). We decided not to discuss viral persistence more in detail as it is not the purpose of this review. 

  1. Line 243: considering the intro specified that the focus of the review would be solely on EBOV, there is a sizeable amount of non-EBOV filoviruses discussed. It might be worth adjusting the intro to be more inclusive to other filoviruses, particularly MARV, as suggested in comments 3 and 9 above.

More information about MARV has been added to the revised manuscript (lines 292-295, 356-361, 568-596).

  1. Line 245-264: possibly change to “majority of other recorded human filovirus infections”; change the semicolon to a comma on line 247; shouldn’t it say 12 years prior on line 252? Revise to “detected in five out of six Gulu survivors” on line 254; delete “despite primary exposure being 11 years ago” as this is redundant what’s written above it in line 252; change to a period after “100 cases” on line 257; change to “sampling patients 9, 15, 21, and 27 months” on line 259; change to “and this diminished by 21 months” on line 263-264.

  1. The sentence has been modified (line 274).
  2. We have carefully checked and were not able to find the semicolon mentioned in this comment.
  3. It has been modified (line 282).
  4. Sentence revised (line 284).
  5. This has been removed.
  6. A full stop was added (line 287).
  7. Revision was made (line 289).
  8. Revision was made (line 293).

  1. Line 267; again, how do these different antibody titers relate to one another, especially when displayed using different metrics/tests?

It is a critical point raised by the reviewer. The use of a WHO research standard can help to compare the neutralisation titres and the neutralisation efficacy between the studies. This point has been added to the manuscript (lines 667-670).

  1. Line 271: possibly change “in which case” to “which suggests that”.

The section has been fully reworded (lines 273-303).

  1. Lines 271-272: does this mean total ab response is more important than neutralizing response? Why might it have lesser importance in protection and what does this mean about nAbs as CoP? Is there any data that relates this difference between MARV and EBOV to MARV’s lack of additional GP products? Finally, would it be useful to briefly discuss studies in the bat reservoir and any antibody responses described for that host, and how that might relate to protection (or not)?

We cannot conclude on the relative importance of neutralising antibody towards protection in MARV as no correlates of protection have yet been defined.  

To our knowledge, there is no comparative data between MARV and EBOV related to the impact of the lack of GP products on antibodies in the context of MARV.

A discussion about antibody responses in bat reservoir may be very interesting but this review focuses on antibody responses in humans and animal models.

  1. The paragraph starting on line 274 might make more sense appearing at the beginning of section 2.1.3, rather than at the end.

It has been added to beginning of the paragraph (lines 275-279).

  1. Line 277-294: what kind of settings do they occur? Change to “settings that have limited”; underline not needed on line 284; seems like it should be “capable of neutralisation, fewer plaques” on line 289; remove the comma after “experiments” and move it to after “organism” on line 290; briefly define how BSL2 relates to BSL4 on line 293; change to “throughput” on line 294.
  1. The settings have been detailed (Line 276-277)
  2. The whole paragraph has been reworded (lines 600-618)
  3. Revision has been added (line 603).
  4. We have specified BSL4. We haven’t considered as essential to specify BSL2 (lines 614-618).
  5. The sentence this relates to has been reworded.

  1. Lines 295-315: move comma from “such as” to after “variables”; add “that” before “the VSV luciferase”; remove “2017” from line 301 and “a” from “has very high correlation to the PRNT”; add a comma after “However” in line 306; paragraph on line 309 should be part of the paragraph above it and not a new one, and it would make more sense to start the sentence as “Another limitation”.

  1. Sentence has been reworded (lines 628)
  2. It has been added to the sentence (line 631).
  3. 2017 has been removed (line 633).
  4. Sentence has been reworded (line 636).
  5. This section has been extensively modified

  1. Lines 321-332: this paragraph goes through mechanisms of neutralization, but it mentions them out of order with when that process occurs during cell entry; it would be clearer and flow better to reorganize this in the order of how the events naturally proceed; on line 325, change to “Further, EBOV fuses”; remove the comma after “however” in line 326; fix the formatting on line 327 for GP2.

The sentence has been reworded (lines 680-682).

  1. Line 341: add “and” before “this is bolstered when”.

The entire paragraph has been reworded (line 694).

  1. Line 343: introduce IgM and IgA antibodies a little better and explain a little more detail about them.

Only a few studies analysed IgM and IgA responses in the context of EBOV. Most studies focused on total IgG. Indeed, the roles of IgM and IgA in neutralisation can be important. We have added this point to the manuscript (lines 187-193). We also added a study which analysed IgA in EVD survivors (lines 257-259).

  1. For Fig 2, the images should be better linked to the text to help illustrate each type of mechanism; within the figure, labeling cellular and viral factors involved in each diagram would be nice to see and aide in clarity; in the legend, clarify the sentence “as well as some methods…” which is currently confusing.

Figure 2 is now Figure 3. More links between the text and figure have been made (lines 675-715).

Too much information within the figure would impact on its clarity. We decided not to change this.

  1. Lines 354-359: not sure if this needs to say “assays” here; change to “measured”; in line 356, remove the period after “IFL” and add “that” before “provided statistically”; add period after “protection” and start a new sentence with “All of the same”; briefly explain what this Spearman’s rank value means in this context; lines 356-359 seem redundant with repeated info in line 356 vs line 358-359.

  1. The word “assay” has been removed (line 739).
  2. Full stop has been removed after IFL (line 734).
  3. That has been added between found and antibodies (line 734).
  4. Full stop has been added after protection (line 735).
  5. Sentence structure has been adopted (line 735).
  6. The definition has been added (line 738).
  7. The paragraph has been restructured to avoid redundancy.

  1. Line 369: the paragraph beginning here should be a part of the previous paragraph as it is a continuation of the same train of thought; add commas after “neutralisation” and after “binds” on line 371.

  1. Now the paragraphs have been merged as suggested (line 741).
  2. Commas added (line 743).

  1. Explain briefly why “antibody-dependent enhancement” on line 392 might not be a good thing in the context of EBOV infection.

A brief explanation of the consequences of ADE has been added (lines 718-720).

  1. Line 399: it might be useful to give generic ab names for Inmazeb and Ebanga in parentheses here.

The trade names have been added in parentheses (line 307).

  1. Line 401: no need for first use of “specific” here.

The paragraph has been reworded (lines 309-314).

  1. Line 407: might be better to stick with use of “Ebanga” here rather than switching to “mAb114”; this would also help prevent those numbers from being right up against “24.2 days”.

A decision has been made to standardise the term to the generic mAb name throughout the review.

  1. Lines 409-410: add comma after “necessary”; pluralize “vaccine”; briefly mention in text what those vaccines are, who makes them, etc., as it’s not clear and readers haven’t gotten to Fig 3 yet

The section has been reworded.

  1. Is there anything that can be done about the massive white space on page 10 and/or the two pages that Fig 3 takes up (which seems like a waste of space given the relatively little text used within the figure).

The layout has been re-arranged.

  1. In Fig 3, clarify that “(cocktail)” means that the name of the larger multi-Ab combination will be in those parentheses; for “Critical residues” clarify that these are GP residues; it might be clearer if the authors included arrows to the regions of GP the residues correspond to as a visual aide to the description; starting with Odesivimab down, why are there no residues of any kind listed, especially since regions of importance are still indicated in the description box? If they’re not known, indicate as such. On page 12, in the description box change to “binds to the Glycan Cap” and add a semicolon after “Cap”. Add a comma after “NHPs” in the CA45 description box; might be worth mentioning if this has been used clinically or not. In FVM04 description box, add period after “NPC1”, delete “hence is” and capitalize “Neutralises”; add comma after “BDBV” and change to “protective”. MR191 should be defined; why is this mAb not described on its own in the Fig?

  1. The whole section dedicated to monoclonal antibodies has been restructured (lines 305-376).
  2. The authors do not believe any clarification is needed given it is in parentheses in the heading.
  3. All mAbs discussed target the GP and all domains have been discussed previously.
  4. The highlighted regions of the GP show where the mAbs bind and arrows may clutter and confuse the diagrams.
  5. Thank you for pointing this out this was a copying error that has been amended
  6. Changed to GC
  7. Sentence has been restructured
  8. Sentence has been restructured
  9. Very few mAbs have been used clinically to the authors knowledge this only extends to the those that participated in the PALM trial furthermore references to this are found in the text
  10. Changes to the FVM04 box have been made
  11. MR191 has been discussed in much more detail in the text (lines 358-363)
  12. MR191 is not in figure as it binds to MARV which has a substantially different structure to EBOV and is not something we have described previously.

  1. Line 415-416: clarify again that these are GP residues.

It has been clarified (line 390).

  1. Lines 423-432: briefly define the PALM trial, what “large” means, where/when the trial took place, etc.; remove comma after “however” on line 427; explain what this post exposure effect was, i.e. when were these patients presenting? Was it soon after exposure, or when symptomatic? If the latter, how sick in general were they? How much lower exactly than ZMapp and Remdesivir? How effective exactly for patients treated early? Give absolute numbers, not just relative ones. Remove comma after “mortality” on line 431.

  1. PALM trial is defined as The Pamoja Tulinde Maisha (PALM [“Together Save Lives” in the Kiswahili language]) trial (line 320).
  2. The size of the trial has been added (line 368).
  3. The location and time of the outbreak has been added (line 366).
  4. Sentence has been reworded.
  5. Mortality was the measured endpoint.
  6. Patients presented following infection and during various stages of disease (line 366-367).
  7. The percentages for ZMapp and Remdesivir have been added (line 370).
  8. This data was not provided in the study.
  9. All arms were run relative to each other.
  10. Comma has been removed (line 370).

  1. Line 435-441: name the drugs again for clarity; revise to “EBOV-specific mAbs, with no licensed”; add a comma after “mutate” on line 439; briefly define the concept of immune escape on line 440, as well as what KZ52 is within the text; doesn’t it also neutralize RESTV?

  1. The paragraph has been restructured and reworded.
  2. Defined the concept of immune evasion (lines 758-761)
  3. KZ52 has been described previously
  4. The section has been reduced to focus on protection.

  1. Lines 444-458: remove the comma after “FVM04”; change to “MARV” on line 448; change to “driver” on line 450 and remove “However” from beginning of sentence; start a new sentence after “ELISA” on line 454; add a comma after “filoviruses” on line 455; on line 456, was flow cytometry used to identify B cells were GP-specific? Is the 0.06% of B cell cross-reactivity enough to protect?

  1. The paragraph has been restructured and reworded.
  2. MARV corrected
  3. Full stop has been added after ELISA and comma added after filoviruses (lines 793-794)
  4. This sentence has been removed to focus on the take-home message of this publication.

  1. In section 5.2, describe how well these multi-Ab cocktails protects (particularly the one with MR191), when they’re given, the regimen, any potential downsides, and address why, if they work very well on multiple filoviruses, they are not being used clinically, or haven’t been licensed or approved.

  1. More details on MR191 have been added in the immune escape section (lines 780-782) and the section on mAbs (lines 358-363).
  2. The review is not really focused on the clinical development of mAbs but more the mechanisms of neutralising antibodies and how they relate to protection.

  1. Line 468-469: add “be” before “administrated”; it might be cleaner to begin a new sentence after “patients” and start the new sentence with “Contaminants include”; add comma after “hepatitis A”.

This section 3.2 has been greatly modified and expanded. The conclusions have been refocused to highlight how neutralising antibodies contribute to convalescent plasma.

  1. Line 475: what are some of the parameters involved?

This is no longer in the newly focused conclusion

  1. Lines 476-480: this section is confusing; rewrite for better clarity; add a comma after “protection” in line 479; what are some hypothetical examples of this “much more varied set” of correlates?

This section dedicated to Convalescent plasma has been fully rewritten. The particular sentence mentioned by the reviewer has been removed.

  1. Line 483-499: you might consider removing “viral vector” here as the description is too cumbersome, and more “vector” after “VSV virus” on line 484. Go into more detail on the Guinea trials, including when they occurred, who was involved, etc. Remove the comma after “only” on line 489, change “but” to “and” on line 495, move “however” to the beginning of the sentence on line 496. What do you mean by “the follow-up of vaccines”, is this supposed to be “vaccinees”? What was the actual experimental bias the authors refer to?

  1. This description has been changed (line 491).
  2. Far more detail has been added about the Erbveo vaccine including the trials (lines 492-521).
  3. The whole section has been extensively modified.
  4. The bias has been explained (line 504-507).

  1. Line 505: rephrase as “post-vaccination, though two out of six NHPs died post-challenge, suggesting” for better clarity.

Sentence rephrased (lines 485-490).

  1. In section 5.4, how efficacious is Zabdeno and Mvabea? Is it protective, and if so, by how much? If this is known information, add these details to the text.

Zabdeno and Mvabea has been through clinical trials for safety and immunogenicity but to the authors knowledge there have been no efficacy or trials to determine protection which can only be postulated by bridging with NHPs pre-clinicial studies.

  1. Lines 508-514: clarify that Zabdeno and Mvabea is a two-phase vaccine candidate, fix spelling of Johnson&Johnson, add commas after “Mvabea and “Johnson” on line 508, add a comma after “vaccine” on line 513, change “and therefore, has the” to “with the”; on line 514, pluralize “glycoprotein”, remove commas after “from” and “SUDV” and add “and” before MARV and change “of TAFV” to “from TAFV”.

  1. This has been clarified (line 538).
  2. Spelling has been corrected (line 539).
  3. Comma has been added (line 537).
  4. This has been reworded (line 544).

  1. Lines 515-516: this is somewhat redundant information to that in lines 510-512. Thus, the authors could rephrase this line as “The order and timing regimen used for these two vaccines was determined by immunogenicity data from phase II trials.”

The sentence was reworded.

  1. Line 518: when is 93% seropositivity achieved?

Time of seropositivity has been added (line 550).

  1. Lines 518-522 are a bit confusing and should be rewritten for better clarity; why was the trial halted?

  1. Sentence was modified (lines 549-553)
  2. We meant the final timepoint of the trial. The sentence has been clarified (line 553)

  1. Line 527: move “with Erbevo” after “vaccination” so it doesn’t sound like the challenge itself was done with Erbevo.

The suggestion has been adopted (line 564).

  1. Line 529: what is MIL-77? Why is this not listed in Fig 3? This should be described more.

The ZMapp cocktail was formed from the merging of two cocktails. MIL77 was an antibody from one of these two cocktails that wasn’t chosen and hasn’t progressed since, hence we considered it was not relevant to include it in Figure 3 and describe it more in this review.

  1. Lines 541-549: add commas after “antibodies” on line 540, after “mAb” on line 542, after “escape” on line 543, and after “Worryingly” on line 546; change “and instead” to “which instead” on line 542; remove comma after “P330S” and add “and”; clarify that KZ52 is a mAb and move the description of this standard to the section before it when KZ52 was first mentioned; in line 546, what are the overlapping epitopes and what are some examples of the other mAbs? Fix “observed” on line 549.

  1. Commas added where appropriate in the new sentence structure (lines 762-765)
  2. Sentence has been removed
  3. Description of KZ52 has been moved (line 328)
  4. Some examples of mAbs with overlapping epitopes have been added (line 768)
  5. “Observed” has been fixed (line 795)

  1. Lines 555-566: add comma after “antibodies”, add “and” before “in particular” and remove comma after that, add comma after “infection” on line 556, add comma after “EVD” on line 560, add comma after “outbreak” on line 562, pluralize “measure” on line 563, add comma after “limited” on line 564, depluralize “mAbs” on line 565, consider changing “treatments” on line 565/566 to “therapeutics” to avoid redundancy of using the same word twice in the sentence.

The conclusion has been extensively modified (lines 802-813)

  1. Line 556: why does the current research fall short exactly? Explain further.

A further explanation has been provided (lines 802-805).

  1. Line 563: is “program” here referring to PALM? If so, clarify.

Program refers to the vaccination drive following the Guinea 2021 outbreak but this has been cut from the revised manuscript.

  1. Line 566: the mention of “better access” seems to emphasize logistics as an important element in antibody treatment, which it is, but this concept of logistics and drug availability is not well emphasized throughout the rest of the paper. This could be corrected through addressing comments 12 and 13 above.

Whilst drug availability is a very valid point and a huge concern, the review is more focused on antibodies. The focus of this review is to understand if neutralising antibodies provide protection from filovirus infection.

Reviewer 3 Report

In the article entitled “Filovirus Neutralising Antibodies: Mechanisms of Action and Therapeutic Application,” Hargreaves and colleagues set out to review the role of neutralizing antibodies in immunity against filovirus infections. This is a very interesting topic in the field, with a lot of data to cover. To this reviewer’s knowledge, there exists no other comprehensive review on this subject, so, in principle, this review is very welcome. Unfortunately, the article possesses many deficiencies. In general, it is poorly written and not well organized. Important topics, such as therapeutic monoclonal antibodies are covered superficially, with key findings/reports left out entirely. Other parts of the paper seem irrelevant or are included with little to connect them to the overall themes of the paper. Below are several comments that should be addressed.

Major Comments:

  1. Filovirus nomenclature is used incorrectly throughout the manuscript. For example:

- Line 15: “EBOV” is the abbreviation for Ebola virus, which is the type virus belonging to the species Zaire ebolavirus. It is therefore not accurate to define “EBOV” as the abbreviation for Zaire ebolavirus.

- Line 29 and 34: The single noun “genus” should be used in place of the plural “genera”.

- Line 29-30: The genus Marburgvirus is composed of one species (Marburg marburgvirus) which contains two viruses, Marburg virus and Ravn virus. Please correct.

- Lines 29, 34, 44, 45, 52, 53, 61: Genus and species names should be italicized.

- Lines 34-36: The abbreviations for each ebolavirus (EBOV, SUDV, BDBV, etc) refer to the virus names (Ebola virus, Sudan virus, Bundibugyo virus, etc) and not the species names, as indicated in this sentence. Please revise.

- Line 40: The abbreviation “EBOV” should not be italicized.

- Line 43: Mengla virus (MLAV) is the type virus of the species Mengla dianlovirus, within the genus Dianlovirus. Please correct.

- Line 45: Lloviu virus (LLOV) is the type virus of the species Lloviu cuevavirus, within the genus Cuevavirus. Please correct.

- Line 47 and elsewhere: It is important to note that species cannot cause disease. Rather, viruses that belong to a specific species cause disease. Please revise throughout.

- Lines 52-53: Huangjiao virus (HUJV) is the type virus of the species Huangjiao thamnovirus, within the genus Thamnovirus. Likewise, Xilang virus (XILV) is the type virus of the species Xilang striavirus, within the genus Striavirus. Please correct.

- Line 244: “BUDV” should be replaced with “BDBV”

- Line 448: Should be “MARV” not “mARV”.

  1. Line 16-17: The authors state that the “… 2013-2016 epidemic in West Africa provided large survivor cohorts which support the concept that specific neutralising antibodies play a key role in protection,” but it is not clear how the existence of survivors, themselves, justifies a role for neutralising antibodies. Please clarify.

  1. Section 1 should be reorganized to include discrete sub-sections that cover genome organization, replication cycle, disease, and immune response. Combining all of these topics into a single sub-section under the heading “Genome organisation of Ebolavirus” makes little sense.

  1. The authors state on Lines 56-58 that they will focus the review on EBOV, yet the title of the manuscript and much of what they discuss covers filoviruses in general. Consider revising.

  1. Lines 153-155: This statement should be clarified to indicate that there is clearly a role for CD8 T cells in animals vaccinated with an adenovirus-vectored EBOV GP vaccine. This is an important distinction because, in a separate study, Marzi et al. demonstrated that CD8 T cell depletion in animals vaccinated with VSV-EBOV all survived virus challenge.

  1. Section 2.1.1: It is not clear why a lengthy discussion of the FDA animal rule and the use of animal models is necessary with respect to “neutralising antibodies in the context of EBOV”. Consider revising or eliminating.

  1. Lines 309: This is an unclear statement. In many cases, EBOV GP-pseudotyped VSV systems used for neutralization assays (or similar assays) are replication competent. The authors are correct in stating that the absence of other filovirus proteins (e.g., VP40) in these systems is a potential disadvantage, but this is not because the system itself is replication deficient.

  1. Section 5: In general, the discussion of monoclonal antibodies is superficial and incomplete. For instance, the authors discuss CA4 and FVM04 efficacy in mice, but do not cite the more recent evaluation in guinea pigs and nonhuman primates. Moreover, there are a number of other neutralizing antibodies (including ADI-15878, BDBV223, EBOV-520, etc.) that aren’t mentioned at all. Considering the amount of research effort that has been dedicated to developing and understanding filovirus monoclonal antibodies, it is reasonable to expect a more thorough review of these data in the present manuscript.

  1. Section 2: The structure of this section should be revised. There exists a sub-section 2.1 with multiple sub-sub-sections, but there is no sub-section 2.1. Further, it is not clear why sections 2.1.1. – 2.1.3. should appear under sub-section 2.1, which focuses specifically on correlates of protection. Finally, it seems counterintuitive to include sub-sub-section 2.1.3 (i.e., “Neutralising antibodies against non-EBOV filoviruses in humans) under the section 2 title, “Neutralising antibodies in the context of EBOV.” Please re-organize.

  1. Section 5.4: It may not be accurate to describe vaccines as “antibody based therapeutics”, although it is important to consider the role of antibodies elicited by these vaccines in conferring protection. Unfortunately, this section offers only a superficial description of the connection between vaccines and neutralizing antibody titers—particularly for VSV-EBOV, for which a lot of data exist. Please consider revising this sub-section, and perhaps moving it to a section of its own. Moreover, Lines 500-507 do not seem relevant and should be removed.

  1. Section 5.5: There exists a significant amount of data regarding immune escape, particularly with respect to mAb treatment. It would be extremely helpful if the authors could review these data in earnest.

  1. In general, citations are poorly placed or missing altogether. Many statements are left unsupported by citations or the appropriate citations are given much later, without a clear connection to the original statement that should have been cited.

  1. The paper should be proofread carefully to correct numerous typos, grammatical errors, and confusing syntax.

Minor Comments:

- Line 36: Assume that the authors mean “The first four of these viruses…”

- Line 45: The paper that first describes the identification of LLOV was published in 2011 (PMID: 22039362). It is not entirely accurate to imply that LLOV was discovered in “2002 and 2017.”

- Lines 64-74: The authors may wish to discuss replication in this section as well.

- Line 89: “, and)” should be deleted.

- Line 101-102: This sentence is confusing and should be revised.

- Line 128: Should be “macropinocytosis.”

- Line 246: Please clarify that the “2000 Gulu outbreak” was caused by SUDV.

- Lines 270-273: Based on the cited study, it’s not clear how the authors can conclude that the neutralising antibody response to MARV wanes more rapidly after infection. Please clarify.

- Line 366-368: It would be more accurate to state something like, “antibodies to the glycan cap, where most attachment factors bind…” Additionally, the sentence implies that antibodies against attachment factors would be needed to prevent cellular entry, but this is likely not the intended meaning. It would be accurate to state, instead, that antibodies capable of binding to the glycan cap and blocking all attachment factors would be needed to prevent entry.

- Section 5.1: It would be helpful if the authors would consistently refer to mAb114 as such, rather than by occasionally using its brand name.

- Line 435: It may be more precise to state the following: “… the only two licensed EBOV-specific mAb therapeutics…” since Inmazeb consists of a cocktail of three mAbs.

- Lines 410-413: This is an important consideration, which is addressed, at least partially, in Lines 526-529. Suggest revising these sections to avoid redundancy and enhance clarity.

- Lines 550-552: This is not relevant. Please delete.

Author Response

In the article entitled “Filovirus Neutralising Antibodies: Mechanisms of Action and Therapeutic Application,” Hargreaves and colleagues set out to review the role of neutralizing antibodies in immunity against filovirus infections. This is a very interesting topic in the field, with a lot of data to cover. To this reviewer’s knowledge, there exists no other comprehensive review on this subject, so, in principle, this review is very welcome. Unfortunately, the article possesses many deficiencies. In general, it is poorly written and not well organized. Important topics, such as therapeutic monoclonal antibodies are covered superficially, with key findings/reports left out entirely. Other parts of the paper seem irrelevant or are included with little to connect them to the overall themes of the paper. Below are several comments that should be addressed.

The authors thank the reviewer for the comments. We extensively worked on the structure of the review in order to improve its clarity. The structure has been modified and some sections were merged. The sections about neutralising responses following natural infection, monoclonal antibodies, vaccines have been expanded as suggested.

Major Comments:

  1. Filovirus nomenclature is used incorrectly throughout the manuscript. For example:

- Line 15: “EBOV” is the abbreviation for Ebola virus, which is the type virus belonging to the species Zaire ebolavirus. It is therefore not accurate to define “EBOV” as the abbreviation for Zaire ebolavirus.

- Line 29 and 34: The single noun “genus” should be used in place of the plural “genera”.

This was modified throughout the manuscript.

- Line 29-30: The genus Marburgvirus is composed of one species (Marburg marburgvirus) which contains two viruses, Marburg virus and Ravn virus. Please correct.

- Lines 29, 34, 44, 45, 52, 53, 61: Genus and species names should be italicized.

- Lines 34-36: The abbreviations for each ebolavirus (EBOV, SUDV, BDBV, etc) refer to the virus names (Ebola virus, Sudan virus, Bundibugyo virus, etc) and not the species names, as indicated in this sentence. Please revise.

It was modified throughout the manuscript.

- Line 40: The abbreviation “EBOV” should not be italicized.

It was modified throughout the manuscript.

- Line 43: Mengla virus (MLAV) is the type virus of the species Mengla dianlovirus, within the genus Dianlovirus. Please correct.

- Line 45: Lloviu virus (LLOV) is the type virus of the species Lloviu cuevavirus, within the genus Cuevavirus. Please correct.

- Line 47 and elsewhere: It is important to note that species cannot cause disease. Rather, viruses that belong to a specific species cause disease. Please revise throughout.

- Lines 52-53: Huangjiao virus (HUJV) is the type virus of the species Huangjiao thamnovirus, within the genus Thamnovirus. Likewise, Xilang virus (XILV) is the type virus of the species Xilang striavirus, within the genus Striavirus. Please correct.

- Line 244: “BUDV” should be replaced with “BDBV”

Thank you for the clarification. The nomenclature has been modified throughout the whole manuscript. All recommended modifications related to the phylogeny of filoviruses have been made with a particular emphasis on clarifying the difference between a species and the viruses belonging to the species.

- Line 448: Should be “MARV” not “mARV”.

The sentence has been modified and all capitalised abbreviations have been checked (Line 795)

  1. Line 16-17: The authors state that the “… 2013-2016 epidemic in West Africa provided large survivor cohorts which support the concept that specific neutralising antibodies play a key role in protection,” but it is not clear how the existence of survivors, themselves, justifies a role for neutralising antibodies. Please clarify.

The sentence in the abstract has been clarified. We specified how the large cohorts enabled a lot of research in the field (Line 16). In addition, the whole abstract has been improved.

  1. Section 1 should be reorganized to include discrete sub-sections that cover genome organization, replication cycle, disease, and immune response. Combining all of these topics into a single sub-section under the heading “Genome organisation of Ebolavirus” makes little sense.

 As suggested by the reviewer, the section 1 has been reorganised with the following sub-sections: “Filoviridae phylogeny”, “Genome Organisation of Ebolavirus”, “Cellular entry of Ebola virus” and “EVD and immunity”.

  1. The authors state on Lines 56-58 that they will focus the review on EBOV, yet the title of the manuscript and much of what they discuss covers filoviruses in general. Consider revising.

The manuscript covers filovirus in general and mentions some studies related to non-EBOV filovirus. However, most studies which dissected immune responses post-infection, post-treatment or post-vaccination were carried out in the context of EBOV. So, the manuscript mainly covers the literature related to EBOV. We revised the sentence to clarify this point (Lines 60-62).

  1. Lines 153-155: This statement should be clarified to indicate that there is clearly a role for CD8 T cells in animals vaccinated with an adenovirus-vectored EBOV GP vaccine. This is an important distinction because, in a separate study, Marzi et al. demonstrated that CD8 T cell depletion in animals vaccinated with VSV-EBOV all survived virus challenge.

The role of T cell responses in animal models post-vaccination have been extensively discussed in a review dedicated to animal models used in EBOV research (Longet et al 2020. 10.3389/fimmu.2020.599568.). In the current review, we focused on the role of humoral response post-vaccination. As suggested by the reviewer, the study Marzi et al. has been included in the manuscript (Lines 451-452).

  1. Section 2.1.1: It is not clear why a lengthy discussion of the FDA animal rule and the use of animal models is necessary with respect to “neutralising antibodies in the context of EBOV”. Consider revising or eliminating.

This sentence was unclear and we decided to delete it.

  1. Lines 309: This is an unclear statement. In many cases, EBOV GP-pseudotyped VSV systems used for neutralization assays (or similar assays) are replication competent. The authors are correct in stating that the absence of other filovirus proteins (e.g., VP40) in these systems is a potential disadvantage, but this is not because the system itself is replication deficient.

Indeed, the statement was unclear. The limitations of the pseudovirus system have been clarified. The whole paragraph has been reorganised to improve its accuracy (Lines 619-670).

  1. Section 5: In general, the discussion of monoclonal antibodies is superficial and incomplete. For instance, the authors discuss CA4 and FVM04 efficacy in mice, but do not cite the more recent evaluation in guinea pigs and nonhuman primates. Moreover, there are a number of other neutralizing antibodies (including ADI-15878, BDBV223, EBOV-520, etc.) that aren’t mentioned at all. Considering the amount of research effort that has been dedicated to developing and understanding filovirus monoclonal antibodies, it is reasonable to expect a more thorough review of these data in the present manuscript.

The section dedicated to monoclonal antibodies has been entirely revised and expanded. We mentioned ZMapp, Inmazeb®, mAb 114, CA45, FVM04, ADI-15878, BDBV223, EBOV-520 and described them in more detail (Lines 305-377).

  1. Section 2: The structure of this section should be revised. There exists a sub-section 2.1 with multiple sub-sub-sections, but there is no sub-section 2.1. Further, it is not clear why sections 2.1.1. – 2.1.3. should appear under sub-section 2.1, which focuses specifically on correlates of protection. Finally, it seems counterintuitive to include sub-sub-section 2.1.3 (i.e., “Neutralising antibodies against non-EBOV filoviruses in humans) under the section 2 title, “Neutralising antibodies in the context of EBOV.” Please re-organize.

The whole review and particularly the section 2 have been fully reorganised.

  1. Section 5.4: It may not be accurate to describe vaccines as “antibody-based therapeutics”, although it is important to consider the role of antibodies elicited by these vaccines in conferring protection. Unfortunately, this section offers only a superficial description of the connection between vaccines and neutralizing antibody titers—particularly for VSV-EBOV, for which a lot of data exist. Please consider revising this sub-section, and perhaps moving it to a section of its own. Moreover, Lines 500-507 do not seem relevant and should be removed.

The section dedicated to the role of neutralising antibodies following vaccination has been expanded and reorganised. More studies have been included. We improved the connection between the results of the studies and the role of neutralising antibodies post-vaccination (Lines 440-568). The irrelevant lines have been deleted.

  1. Section 5.5: There exists a significant amount of data regarding immune escape, particularly with respect to mAb treatment. It would be extremely helpful if the authors could review these data in earnest.

We decided to merge the sections dedicated to immune escape with the strategies to prevent immune escape. This section has been fully revised (Lines 658-799).

  1. In general, citations are poorly placed or missing altogether. Many statements are left unsupported by citations or the appropriate citations are given much later, without a clear connection to the original statement that should have been cited.

The structure of the whole manuscript has been revised. The connection between the statements and the citations has been improved.

  1. The paper should be proofread carefully to correct numerous typos, grammatical errors, and confusing syntax.

The manuscript has been proofread by several people.

Minor Comments:

- Line 36: Assume that the authors mean “The first four of these viruses…”

Yes, the sentence has been modified as suggested (Line 41).

- Line 45: The paper that first describes the identification of LLOV was published in 2011 (PMID: 22039362). It is not entirely accurate to imply that LLOV was discovered in “2002 and 2017.”

The year of discovery was corrected. The new reference has been added. (Line 47).

- Lines 64-74: The authors may wish to discuss replication in this section as well.

Replication of EBOV is very interesting but we decided not to discuss replication as it is not the topic of this review.

- Line 89: “, and)” should be deleted.

It was deleted (Line 97).

- Line 101-102: This sentence is confusing and should be revised.

The sentence has been revised (Lines 111-115).

- Line 128: Should be “macropinocytosis.”

The word was modified (Line 133).

- Line 246: Please clarify that the “2000 Gulu outbreak” was caused by SUDV.

It was clarified that SUDV was causative agent of the 2000 outbreak (Line 281).

- Lines 270-273: Based on the cited study, it’s not clear how the authors can conclude that the neutralising antibody response to MARV wanes more rapidly after infection. Please clarify.

The paragraph has been clarified (Lines 294-297).

- Line 366-368: It would be more accurate to state something like, “antibodies to the glycan cap, where most attachment factors bind…” Additionally, the sentence implies that antibodies against attachment factors would be needed to prevent cellular entry, but this is likely not the intended meaning. It would be accurate to state, instead, that antibodies capable of binding to the glycan cap and blocking all attachment factors would be needed to prevent entry.

The sentence has been fully revised (Lines 747-749).

- Section 5.1: It would be helpful if the authors would consistently refer to mAb114 as such, rather than by occasionally using its brand name.

It has been modified throughout the review.

- Line 435: It may be more precise to state the following: “… the only two licensed EBOV-specific mAb therapeutics…” since Inmazeb consists of a cocktail of three mAbs.

The sentence has been modified (Line 776).

- Lines 410-413: This is an important consideration, which is addressed, at least partially, in Lines 526-529. Suggest revising these sections to avoid redundancy and enhance clarity.

The sections have been revised to improve the clarity.

- Lines 550-552: This is not relevant. Please delete.

This sentence related to SARS-CoV-2 has been deleted.

Round 2

Reviewer 3 Report

The authors have substantially modified and improved their review in response to both sets of reviewer comments. The resulting manuscript is much more thorough, well organized, and impactful. This reviewer has only a few minor suggestions, as follows:

  1. Filovirus nomenclature is still used incorrectly or inconsistently. Suggest consulting the following publications for guidance on how to improve: PMIDs 31021739, 28653188, 30926957

  1. The manuscript should be carefully reviewed to correct a number of typos and grammatical errors. For example,

- Line 48: “Lloviu” instead of “LLoviu”.

- Lines 59, 65: “Thamnovirus” instead of “Thamnorvirus”.

- Line 276: “have been often occur” should be revised.

- Line 306: delete apostrophe after “EBOV”.

- Line 356: “It has ben shown protection” should be revised.

- Line 359: Include “mAb” after “RAVV”?

- Line 361: Include “binding” after “receptor”?

- Line 772: Typo: “increas”.

  1. For EBOV, it is co-translational editing of the GP transcript (i.e., transcriptional slippage) that results in the addition of non-templated adenosines to the GP gene, resulting in alternate products. Thus, in Line 84 it may be redundant to say “transcriptional slippage and co- … translational editing.” Further, in Lines 85-86, the authors may want to specify that similar co-translational and post-translational modifications are not observed in MARV GP (rather than simply stating that “post-translational modifications are not observed in MARV,” which is vague). Finally, it may be worth specifying the number of non-templated adenosines that are (or can be) added to the GP gene, rather than just stating “adenosines.”

  1. Lines 445-447: This sentence seems contradictory.

  1. Lines 652-655: As described here, a chimeric virus is not much different from the pseudotyped viruses described above. The authors should clarify this paragraph, and may consider focussing on chimeric filoviruses, in particular, rather than chimeric viruses, in general.

Author Response

The authors have substantially modified and improved their review in response to both sets of reviewer comments. The resulting manuscript is much more thorough, well organized, and impactful. This reviewer has only a few minor suggestions, as follows:

  1. Filovirus nomenclature is still used incorrectly or inconsistently. Suggest consulting the following publications for guidance on how to improve: PMIDs 31021739, 28653188, 30926957

The authors thank for the publications. The filovirus nomenclature has been corrected and clarified (lines 31-64).

  1. The manuscript should be carefully reviewed to correct a number of typos and grammatical errors. For example,

Thanks, the revised manuscript has been reviewed.

- Line 48: “Lloviu” instead of “LLoviu”.

It has been corrected (line 48).

- Lines 59, 65: “Thamnovirus” instead of “Thamnorvirus”.

The word has been modified (lines 59, 66).

- Line 276: “have been often occur” should be revised.

It has been revised (line 279).

- Line 306: delete apostrophe after “EBOV”.

Apostrophe has been deleted (line 309).

- Line 356: “It has ben shown protection” should be revised.

It has been revised (line 359).

- Line 359: Include “mAb” after “RAVV”?

The word mAb has been added (line 362).

- Line 361: Include “binding” after “receptor”?

It has been included (line 364).

- Line 772: Typo: “increas”.

It has been corrected (line 780).

  1. For EBOV, it is co-translational editing of the GP transcript (i.e., transcriptional slippage) that results in the addition of non-templated adenosines to the GP gene, resulting in alternate products. Thus, in Line 84 it may be redundant to say “transcriptional slippage and co- … translational editing.” Further, in Lines 85-86, the authors may want to specify that similar co-translational and post-translational modifications are not observed in MARV GP (rather than simply stating that “post-translational modifications are not observed in MARV,” which is vague). Finally, it may be worth specifying the number of non-templated adenosines that are (or can be) added to the GP gene, rather than just stating “adenosines.”

The paragraph has been clarified and the number of adenosines added to the GP gene has been mentioned (lines 85-101).

  1. Lines 445-447: This sentence seems contradictory.

The sentence has been revised (line 451).

  1. Lines 652-655: As described here, a chimeric virus is not much different from the pseudotyped viruses described above. The authors should clarify this paragraph, and may consider focussing on chimeric filoviruses, in particular, rather than chimeric viruses, in general.

The paragraph has been clarified and focused on chimeric filoviruses (lines 656-663).